# Structure of the DDB1-AMBRA1 E3 ligase receptor complex linked to cell cycle regulation

Ming Liu[1], Yang Wang [1], Fei Teng [1,2], Xinyi Mai[1], Xi Wang[1], Ming-Yuan Su [2,3,4] ✉ & Goran Stjepanovic [1] ✉

AMBRA1 is a tumor suppressor protein that functions as a substrate receptor of the ubiquitin conjugation system with roles in autophagy and the cell cycle regulatory network. The intrinsic disorder of AMBRA1 has thus far precluded its structural determination. To solve this problem, we analyzed the dynamics of AMBRA1 using hydrogen deuterium exchange mass spectrometry (HDX-MS). The HDX results indicated that AMBRA1 is a highly flexible protein and can be stabilized upon interaction with DDB1, the adaptor of the Cullin4A/B E3 ligase. Here, we present the cryo-EM structure of AMBRA1 in complex with DDB1 at 3.08 Å resolution. The structure shows that parts of the N- and C-terminal structural regions in AMBRA1 fold together into the highly dynamic WD40 domain and reveals how DDB1 engages with AMBRA1 to create a binding scaffold for substrate recruitment. The N-terminal helix-loop-helix motif and WD40 domain of AMBRA1 associate with the double-propeller fold of DDB1. We also demonstrate that DDB1 binding-defective AMBRA1 mutants prevent ubiquitination of the substrate Cyclin D1 in vitro and increase cell cycle progression. Together, these results provide structural insights into the AMBRA1-ubiquitin ligase complex and suggest a mechanism by which AMBRA1 acts as a hub involved in various physiological processes.

The two major intracellular protein degradation pathways in eukaryotic cells, autophagy and the ubiquitin–proteasome system (UPS), are crucial to maintain cell homeostasis and the regulation of several cellular processes, including protein quality control, cell proliferation and apoptosis[1,2]. Ubiquitin is a highly conserved protein that is most commonly attached to lysine residues in target proteins through the sequential action of ubiquitin-activating enzymes (E1), ubiquitin-conjugating enzymes (E2) and ubiquitin ligases (E3). First, E1 activates the C-terminal glycine of ubiquitin in an ATP-dependent reaction. The activated ubiquitin is then handed over to E2 and finally covalently attached to the substrate protein by E3 ubiquitin ligase. Proteins can be modified by the addition of a single ubiquitin (monoubiquitination) or a chain of multiple ubiquitin molecules (polyubiquitination), which signals the protein for degradation by proteosomes. Cullin-RING ligases (CRLs) are the largest E3 ligase family in eukaryotes and are organized by a scaffold protein Cullin and a catalytic RING subunit, RBX1 or RBX2[3,4]. Damage-specific DNA binding protein 1 (DDB1) is a component of the Cullin4A/B-RING E3 ubiquitin ligase (CRL4) complex and functions as an adaptor protein between Cullin4A/B (CUL4A/B) and CUL4A-associated factors (DCAFs) to target substrates for ubiquitination. DDB1 has triple β-propeller (BPA, BPB and BPC) domains and can associate with diverse substrate receptors, which in turn recruit substrates.

AMBRA1 (activating molecule in Beclin-1 regulated autophagy) is one of the DCAF substrate receptors that plays a central role in the

[1]Kobilka Institute of Innovative Drug Discovery, School of Medicine, The Chinese University of Hong Kong, Shenzhen, Shenzhen 518172, China. [2]Department of Biochemistry, School of Medicine, Southern University of Science and Technology, Shenzhen 518055, China. [3]Key University Laboratory of Metabolism and Health of Guangdong, Southern University of Science and Technology, Shenzhen 518055, China. [4]Institute for Biological Electron Microscopy, Southern University of Science and Technology, Shenzhen 518055, China. ✉e-mail: sumy@sustech.edu.cn; goranstjepanovic@cuhk.edu.cn

communication between autophagy, cell-cycle control and the UPS[5]. AMBRA1 can regulate autophagy at several stages. Under starvation conditions, AMBRA1 promotes the ubiquitination of the serine/threonine kinase ULK1 (Unc-51-like autophagy activating kinase-1) by the E3 ligase TNF receptor associated factor 6 (TRAF6). ULK1 is a component of the multi-subunit ULK complex and is essential for the induction of autophagy. Such ubiquitination results in ULK1 self-association, activation and autophagy initiation[6]. In nutrient-rich conditions, mTORC1 (mammalian target of rapamycin complex 1) phosphorylates AMBRA1 on Ser-52, resulting in the termination of the autophagy response by preventing ULK1 ubiquitination[6,7]. In addition to promoting ULK1 ubiquitination, AMBRA1 functions as an E3 substrate receptor for Beclin1 ubiquitination during starvation-induced autophagy. Beclin1 is a component of the PI3KC3 (class III phosphatidylinositol 3-kinase) complex together with Vps34 lipid kinase responsible for generating PI3P (phosphatidylinositol 3-phosphate). In this process, AMBRA1 promotes the Lys-63 ubiquitination of Beclin1. This modification enhances the interaction of Beclin1 with PI3KC3 and promotes the activation of Vps34, which is required for autophagy induction[8].

Recently, AMBRA1 was identified as the major regulator of the stability of Cyclin D1, which is involved in regulating cell cycle progression. Cyclin D1 interacts with cyclin-dependent kinase 4 and 6 (CDK4/6) to drive cell proliferation by phosphorylating retinoblastoma protein Rb, which in turn dissociates from the E2F transcription factor, resulting in the expression of genes required for S-phase entry. AMBRA1 limits CDK4/6 activity by mediating ubiquitination and degradation of Cyclin D1 as part of the CRL4[DDB1] E3 ligase complex. Loss of AMBRA1 leads to accumulation of Cyclin D1 and decreased sensitivity to CDK4/6 inhibitors, resulting in increased tumorigenic potential[9–11]. Despite the centrality of AMBRA1 in the intersection of autophagy, the UPS and cell growth, the structure and regulatory mechanisms are largely unknown. Here, we determined the cryo-EM structure of the human DDB1-AMBRA1[WD40] complex, which revealed a split WD40 domain formed by the N- and C-terminal AMBRA1 regions. HDX-MS analysis showed that the binding of DDB1 stabilizes distinct AMBRA1 regions, allowing for structural determination. The N-terminal helix-loop-helix motif of AMBRA1 is essential for DDB1 binding, and point mutations within this region abolish their interaction, prevent the ubiquitination of Cyclin D1 in vitro and increase cell cycle progression. Together, these results provide structural insights into the AMBRA1-ubiquitin ligase complex and suggest a mechanism by which AMBRA1 targets proteins involved in essential cell cycles and autophagy pathways.

## Results

### Hydrogen-deuterium exchange analysis of AMBRA1 dynamics

A detailed picture of the mechanism by which AMBRA1 engages CRL4 is needed to provide new insight into AMBRA1 regulation, the relationship between autophagy, the UPS pathway and tumor growth, and for the development of novel therapeutics. We therefore set out to dissect the molecular basis of AMBRA1 recruitment to CRL4. AMBRA1 has been found to contain a WD40 domain in the N-terminal region, while the rest of the protein is proposed to consist of intrinsically disordered regions (IDRs) due to a lack of recognizable protein domains (Fig. 1a)[12]. Hydrogen deuterium exchange mass spectrometry (HDX-MS) can be used to study changes in protein dynamics, binding and folding as well as IDRs or regions with altered conformations between different states[13]. The accessibility of a backbone amide hydrogen is largely dependent on hydrogen bonding and local secondary structure, as well as solvent accessibility[14,15]. Stably folded secondary structure elements such as α-helices or β-sheets incorporate deuterium to a lower degree than interconnecting loops or unstructured regions.

To gain insight into the backbone flexibility of AMBRA1, purified full-length AMBRA1 was analyzed by HDX-MS (Fig. 1b–d). Experiments

were carried out at six time points of amide hydrogen exchange (10, 30, 60, 300, 900, 1800 s). A total of 203 peptides covering 98.8% of the sequence were identified and quantified (Supplementary Fig. 1, Supplementary Table 1). The backbone amide groups in the first 204 residues of the protein exhibit relatively low deuterium uptake, followed by rapid deuteration in regions thought to primarily be intrinsically disordered polypeptide segments (Fig. 1c, Supplementary Fig. 1). In addition to the N-terminal region, the AMBRA1 C-terminal region (residues 853–1044) displayed reduced deuterium uptake, suggesting the formation of a secondary structure. Secondary structure and domain prediction of the N-terminal region indicates the presence of continuous β-strands that fold into three and half WD40 repeats with a length of approximately 40 residues in a single repeat (Supplementary Fig. 2). The WD40 domains can serve as hotspots for protein–protein or protein–DNA interactions and exhibit a β-propeller architecture, most often comprising seven repeats[16]. Therefore, the AMBRA1 N-terminal region alone was unlikely to exhibit a β-propeller architecture. Based on the AlphaFold2 prediction and recent in silico analysis, the N-terminal AMBRA1 is likely to form half of the WD40 domain (WD40-N), and it requires its C-terminal part (WD40-C) to complete the entire β-propeller, connecting by a long loop in between (Supplementary Fig. 3a)[17,18]. The second half of the "split" WD40 domain was mapped to the C-terminal region (residues 853-1044) due to its reduced deuteration levels and the predicted secondary structure consisting of β-strands (Fig. 1c, Supplementary Fig. 1, Supplementary Fig. 3b). HDX-MS revealed multiple peptides spanning the predicted AMBRA1 WD40 domain that exhibited a bimodal isotope pattern in mass spectra (Fig. 1d, Supplementary Figs. 4-5). A bimodal distribution could result from EX1-type HDX kinetics where multiple amide hydrogens exchange in the same conformational fluctuation[13,19]. The EX1 kinetic regime is observed in proteins that undergo cooperative unfolding events that simultaneously expose multiple adjacent amide hydrogens for exchange. Under these conditions, the refolding rate is much lower than the intrinsic hydrogen exchange rate, and all the amide hydrogens exchange with deuterium in the unfolded (open) state before refolding occurs, resulting in two distinct mass envelopes[20,21]. The lower mass envelope represents the closed state, and the higher mass envelope represents the open state.

To identify the CRL4 binding region in AMBRA1, we co-transfected and reconstituted the AMBRA1-CRL4 complex (Fig. 1b). This purified sample was analyzed by HDX-MS, and the data were compared to the deuterated mass spectra for the apo state of AMBRA1. Upon CRL4 complex binding, the lower mass population becomes more prominent relative to the higher mass population in a number of peptides mapping to the WD40 domain of AMBRA1 (Fig. 1d, Supplementary Fig. 3b). This behavior is exemplified by peptides spanning the WD40 domain and predicted IDR regions (Supplementary Figs. 4–5). In the AMBRA1 apo form, the closed state is rapidly converted into a more accessible open state. In contrast, CRL4 complex binding stabilized the closed state and slowed the interconversion between the two states. These changes were consistent with a major stabilization effect on the split WD40 domain dynamics when bound to the CRL4 complex and support its direct role in mediating interaction with E3 ligase adaptors.

### Cryo-EM structure of AMBRA1[WD40] complexed with DDB1

Based on HDX information, a truncated construct comprising residues 1–204 directly fused to residues 853–1044, referred to as AMBRA1[WD40], was designed for cryo-EM structural studies (Fig. 1a, Supplementary Figs. 1 and 3) and then co-expressed with the DDB1 E3 adaptor in HEK 293F cells. The resulting complex contained both subunits at apparently equal stoichiometry, and the subunits co-migrated as a single peak on gel filtration chromatography (Fig. 2a, b). The complex was enzymatically active, as judged by an in vitro pull-down experiment and ubiquitination assay using Cyclin D1 as a substrate (Fig. 3c, d), demonstrating that removal of the AMBRA1

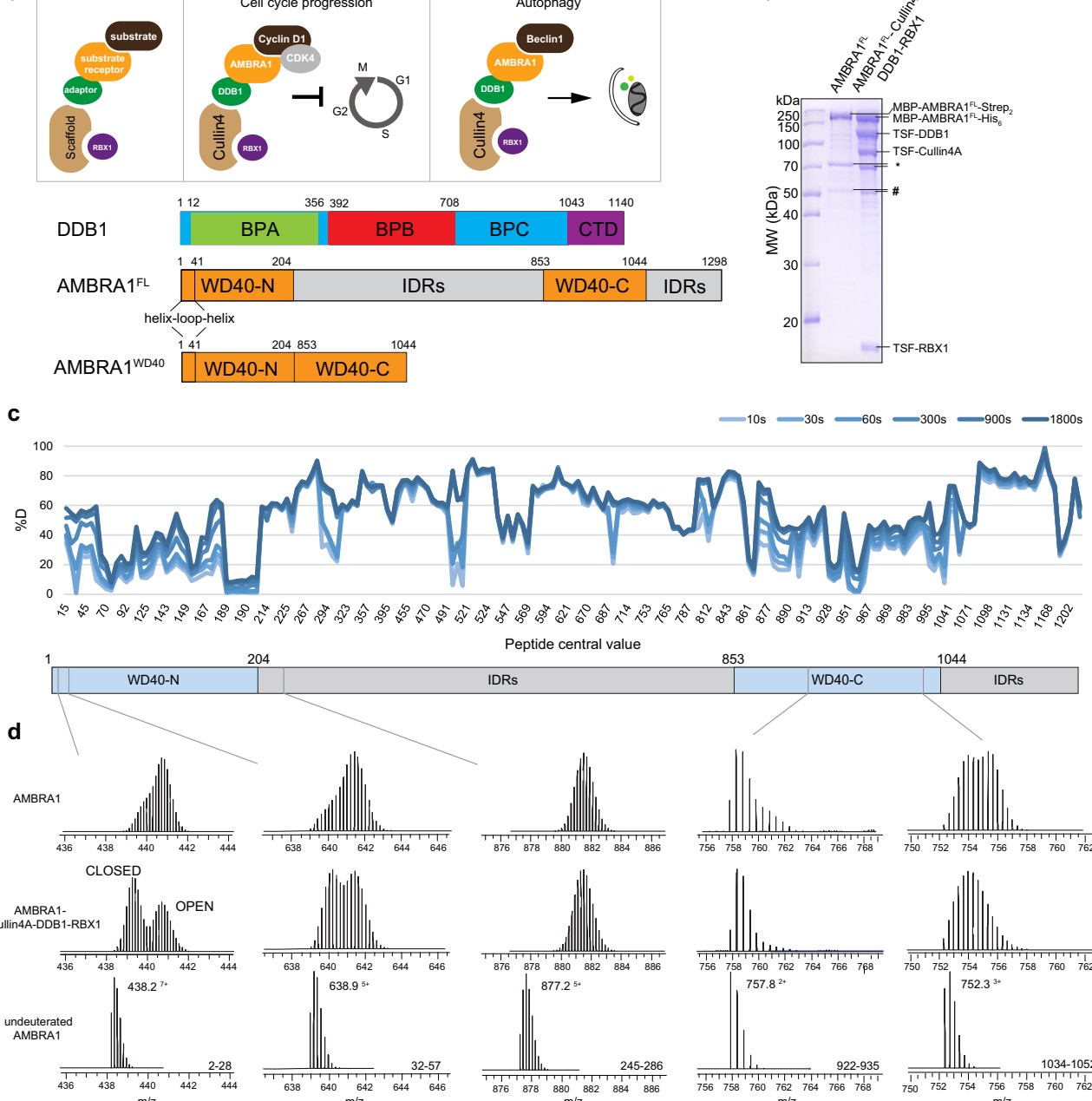

**Fig. 1 | HDX-MS analysis of AMBRA1. a** Cartoon schematic of the function of Cullin4-DDB1-RBX1-AMRBA1 E3 ligase in cell cycle progression and autophagy. Annotated DDB1 and AMBRA1 domain schematics. The AMBRA1$^{WD40}$ construct including the N-terminal helix-loop-helix for cryo-EM was defined. BPA-BPC β-propeller A-C, CTD C-terminal helical domain, FL full length, IDRs intrinsically disordered regions. **b** Coomassie blue-stained SDS-PAGE analysis of purified AMBRA1$^{FL}$ alone and the AMBRA1$^{FL}$-DDB1-Cullin4A-RBX1 complex used for HDX-MS measurements. TSF twin-strep-flag. Strep$_2$ strep-strep. The asterisk and hash indicate hsp70 and other contaminations. MW, molecular weight. The experiment was

repeated independently three times with similar results. Source data are provided as a Source Data file. FL, full length. **c** Plot representing deuterium uptake by full-length AMBRA1, with each data point representing the central residue of an individual peptide. The Y-axis represents % deuteration for a given peptide at each time point. HDX data statistics are given in Supplementary Table 1. IDRs, intrinsically disordered regions. **d** Isotopic envelopes for the selected peptides from AMBRA1, either alone or in complex with Cullin4A-DDB1-RBX1, following 3 s of incubation in D$_2$O.

disordered regions between WD40-N and WD40-C does not disrupt the AMRBA1$^{WD40}$ structure and function of the protein.

We next determined the structure of AMBRA1$^{WD40}$ complexed with DDB1 by cryo-EM. Cryo-EM images were collected and processed as detailed in the "Methods" section (Supplementary Fig. 6). The overall resolution for the structure is 3.08 Å, allowing visualization of the large side chains in the reconstruction (Supplementary Fig. 6d, Supplementary Fig. 7c). The structure of DDB1-AMBRA1$^{WD40}$ showed that 4

individual beta-propeller domains cluster with each other, with the longest dimension of ~130 Å (Fig. 2c). DDB1 has three seven-bladed WD40 β-propellers (referred to as BPA, BPB and BPC) and a C-terminal helical domain (CTD). BPA and BPC pack against each other in the clamp-shaped double-β-propeller fold, forming a large pocket at the interface. There is a low occupancy in DDB1 as determined by particle imaging analyses, and the most flexible part is the BPB domain. 3D classification without alignment analysis demonstrated that this

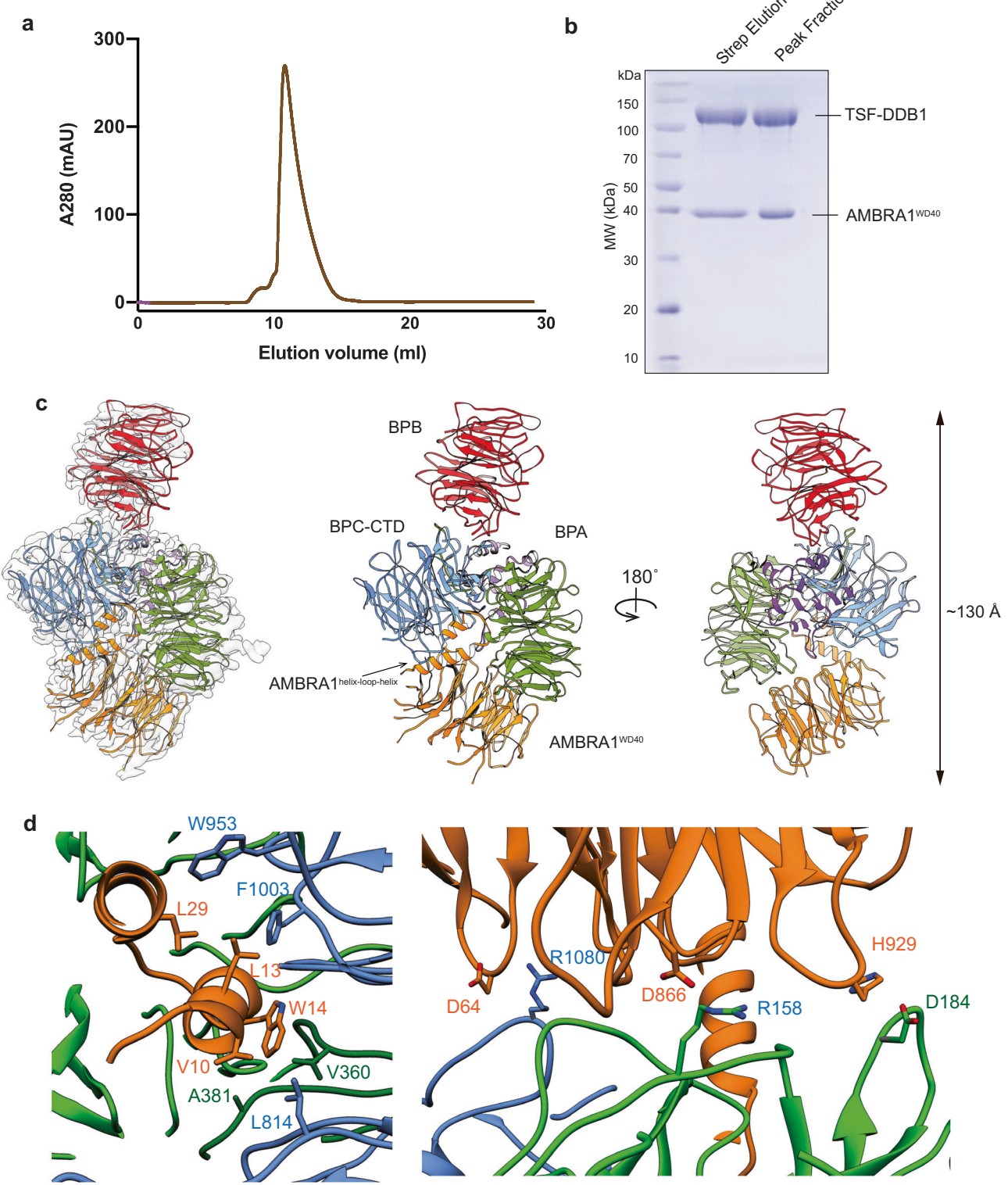

**Fig. 2 | The structural organization of the AMBRA1^WD40-DDB1 complex. a** The size exclusion profile (Superdex 200 Increase 10/300) of the purified AMBRA1^WD40-DDB1 complex. mAU, milliabsorbance units. **b** Coomassie blue-stained SDS-PAGE analysis of the AMBRA1^WD40-DDB1 complex used for structure determination. The left lane was the sample from a strep elution, and the right lane sample was the pooled peak fraction from the size exclusion column. MW, molecular weight. TSF, twin-strep-flag. The experiment was repeated independently three times with similar results. Source data are provided as a Source Data file. **c** The cryo-EM density map and the refined coordinates of the AMBRA1^WD40-DDB1 complex. AMBRA1^WD40 and the DDB1 domain are colored as follows: AMBRA1^WD40, orange; BPA domain, green; BPB domain, red; BPC-CTD domain, cornflower blue and purple. **d** Close-up view of the AMBRA1^WD40-DDB1 interface. The N-terminal helix-loop-helix of AMBRA1 is mainly responsible for the interaction. The key residues contributing to the interface are labeled, and some of those residues are further mutated and confirmed with pull-down experiments.

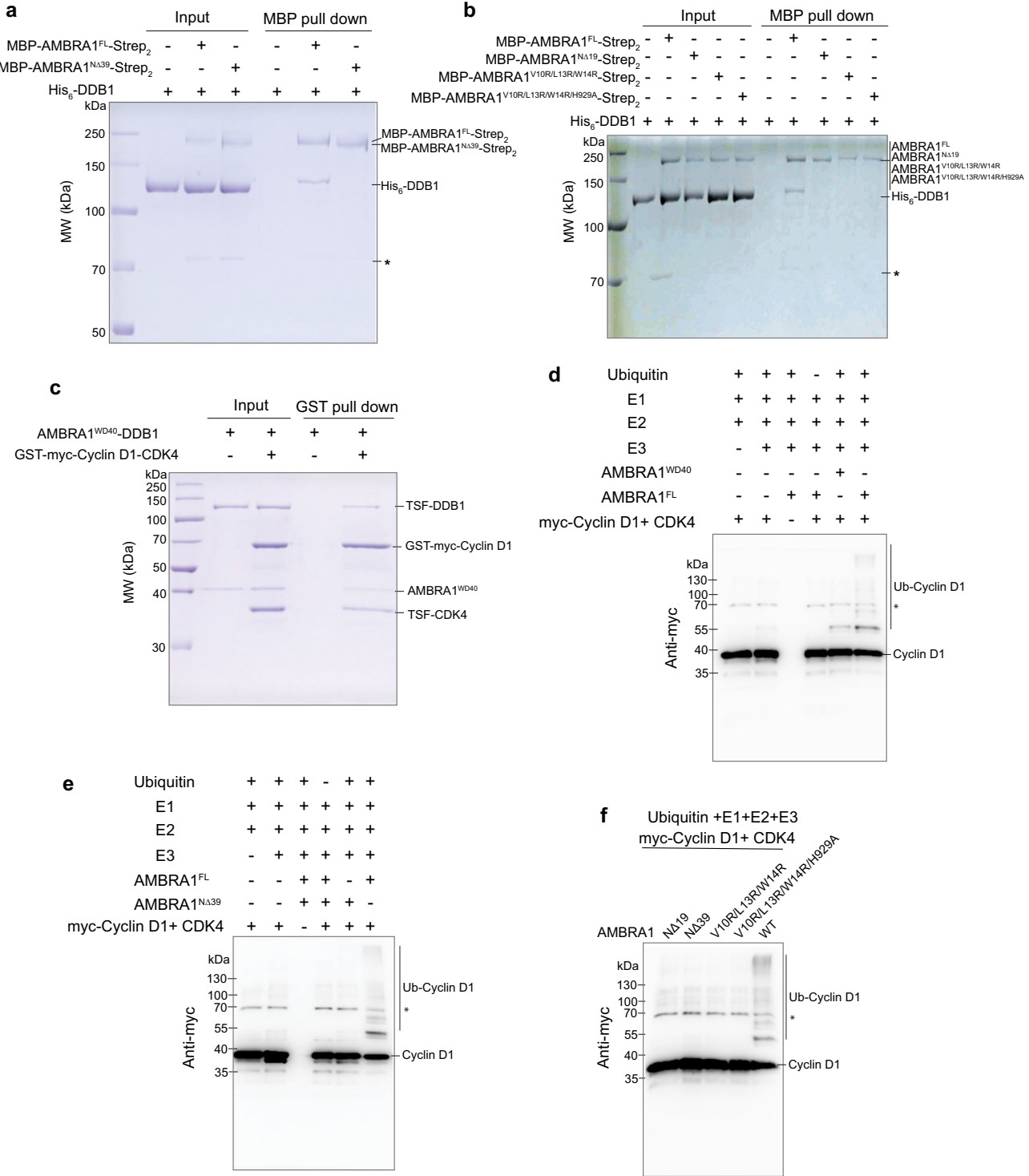

**Fig. 3 | AMBRA1-DDB1-Cullin4A-RBX1-dependent ubiquitination of Cyclin D1. a** In vitro pull-down experiment of full-length and N-terminal 39 residue-deleted AMBRA1 with His$_6$-tagged DDB1. The experiment was repeated at least three times and visualized by SDS-PAGE and Coomassie blue staining. The asterisk indicates hsp70 contamination. FL, full length. MW, molecular weight. Source data are provided as a Source Data file. **b** In vitro pull-down experiment of full-length, N-terminal 19 residue-deleted or mutated AMBRA1 with His$_6$-tagged DDB1. The experiment was repeated at least three times and visualized by SDS-PAGE and Coomassie blue staining. FL, full length. MW, molecular weight. The asterisk indicates hsp70 contamination. Source data are provided as a Source Data file. **c** In vitro pull-down experiment of AMBRA1$^{WD40}$-DDB1 with the Cyclin D1-CDK4 complex. The

experiment was repeated independently three times with similar results. TSF, twin-strep-flag. Source data are provided as a Source Data file. **d** In vitro ubiquitination assay of AMBRA1$^{FL/WD40}$ for Cyclin D1. The asterisk indicates GST-myc-Cyclin D1 from purified myc-Cyclin D1. The experiment was repeated independently three times with similar results. Source data are provided as a Source Data file. **e, f** Ubiquitination assay demonstrated that the N-terminal helix-loop-helix of AMBRA1 is responsible for the polyubiquitination of Cyclin D1. Ubiquitination assays were analyzed by SDS-PAGE, followed by immunoblotting. The asterisk indicates GST-myc-Cyclin D1 from purified myc-Cyclin D1. The experiment was repeated independently three times with similar results. Source data are provided as a Source Data file.

domain can adopt different orientations relative to the BPA-BPC double-β-propeller fold for substrate presentation (Supplementary Fig. 8). This is consistent with previous studies showing that the BPB domain is responsible for interaction with Cullin4 and enables dynamic positioning of the substrates for ubiquitination[22–24].

AMBRA1[WD40] is characterized by an N-terminal helix-loop-helix motif (E6-K41) (Fig. 1a, Supplementary Fig. 4c) and a seven-blade WD40 β-propeller domain that packs against the BPA domain of DDB1[25]. WD40-N adopts Blades I-III and the first two strands of Blade IV, while WD40-C contributes to the latter two strands of Blade IV and the remaining Blades V-VII. Canonical WD40 domains have seven blades or repeats, each of which contains 40–60 residues, which fold into a β-propeller structure[26]. The WD40 domain of AMBRA1 differs significantly from the canonical WD40 domains found in other CRL substrate receptors, such as DCAF1 and CSA. The most prominent difference occurs in blade IV at the transition between the βb and βc strands. In DCAF1, the β-hairpin is formed by the βb and βc strands oriented in an antiparallel direction and linked by a short loop of three amino acids. In AMBRA1, this several-residue long loop is replaced by the 650-residue intrinsically disordered region that separates the WD40-N and WD40-C parts (Supplementary Fig. 9a).

## AMBRA1 binding to the DDB1 WD40 domain

AMBRA1 interacts with DDB1 via a bipartite structure consisting of an N-terminal helix-loop-helix motif and a globular core WD40 domain. Helical extension is commonly found in substrate receptors that bind to DDB1 of the CRL4 E3 ubiquitin ligase. The removal of this helical extension (AMBRA1[NΔ39]) completely abolished the interaction with DDB1, as demonstrated by an in vitro MBP pull-down assay (Fig. 3a). The N-terminal helical motif is almost entirely engulfed by the large pocket formed between BPA and the BPC double propeller of DDB1, stabilizing by the hydrophobic interactions contributed by W953, F1003, V360, L814, and A381 of the BPC domain with V10, L13, and W14 in the first helix of AMRBA1. Additional DDB1-AMBRA1 interactions include the ionic bonding of AMBRA1 D64 with BPC R1080, AMBRA1 H929 with BPA D184, and AMBRA1 D866 with BPA R158 in the loop regions (Fig. 2d). The AMBRA1 residues were tested by mutagenesis by replacement of the hydrophobic residues with the positively charged amino acid arginine. We incubated purified MBP-tagged AMBRA1[FL], AMBRA1[NΔ19] (the helix where V10, L13 and W14 locate), AMBRA1[V10R/L13R/W14R], AMBRA1[V10R/L13R/W14R/H929A] (Fig. 3b) or AMBRA1 single mutants with His₆-tagged DDB1 (Supplementary Fig. 10a). V10R, L13R and W14R point mutants were chosen because they are located within the N-terminal helix-loop-helix motif and mediate interactions between WD40-N and DDB1, while H929A may impact DDB1 interactions with WD40-C (Fig. 2d). When we mutated V10 and L13 to arginine residues, DDB1 binding was completely abolished. Similarly, all combined mutations and N-terminal deletions, including V10 and L13, abolished the interaction of DDB1 with AMBRA1. In contrast, AMBRA1 mutants W14R and H929A did not show any alteration in DDB1 interaction in the in vitro pull-down assay (Supplementary Fig. 10a). Collectively, these results confirmed that the AMBRA1 N-terminal helix-loop-helix is the most central region for DDB1 association.

Comparison of our AMBRA1-DDB1 structure to other DDB1 complexes shows that the binding between DDB1 and AMBRA1[WD40] is similar to other adaptors bound to DDB1, including DCAF1 (PDB: 6ZUE), Simian virus 5 V (PDB: 2B5L) and CSA (PDB: 6FCV), suggesting a common molecular mechanism of DDB1 recognition (Supplementary Fig. 9b). To investigate how AMBRA1 functions in substrate recruitment to the CRL4 E3 ligase machinery, we generated a model of the entire CRL4-AMBRA1 assembly by superimposing the BPB of DDB1 in our structure onto the same domain in the CRL4 E3 structure (PDB 2HYE, Supplementary Fig. 9c). Our model revealed a ~70 Å distance between RBX1 and the central cavity of AMBRA1 (Supplementary

Fig. 9c). Notably, the 3D classification without alignment analysis confirmed the plasticity of the BPB domain of DDB1, which can adopt diverse orientations to position AMBRA1 for substrate ubiquitination. RBX1 is likely to be repositioned as a result of neddylation, therefore establishing an open active conformation to mediate the transfer of ubiquitin from E2 to the substrate, which shortens the distance between the substrates and the enzymes.

## AMBRA1[WD40] can directly interact with its substrate Cyclin D1 and mediate its ubiquitination

AMBRA1 can target Cyclin D1 for ubiquitin-mediated degradation, thereby controlling cell cycle progression through CRL4 E3 ligase[9–11]. To address whether AMBRA1[WD40] is sufficient for substrate recognition and ubiquitination, we purified the Cyclin D1-CDK4 complex for in vitro assays (Supplementary Fig. 11d). Our GST pull-down experiment demonstrated that purified GST-tagged Cyclin D1-CDK4 can associate with the AMBRA1[WD40]-DDB1 complex (Fig. 3c). Furthermore, AMBRA1[WD40] is sufficient to promote ubiquitination of Cyclin D1 but is not as robust as full-length AMBRA1, indicating that the IDR within AMBRA1 may contribute to either augmenting the interactions or other unknown functions (Fig. 3d).

AMBRA1 functions as a substrate receptor to recruit Cyclin D1. Therefore, disrupting the interaction between AMBRA1 and DDB1 would fail to mediate the ubiquitination of its substrate. Indeed, Cyclin D1 is polyubiquitinated if every enzyme for the ubiquitination cascade reaction is included. When we excluded ubiquitin, CRL4 E3 ligase or full-length AMBRA1, we did not observe any ubiquitination. In contrast, AMBRA1[NΔ39] failed to promote the polyubiquitination of Cyclin D1 (Fig. 3e, Supplementary Fig. 12a). We further tested the effects of AMBRA1 mutations, including V10R, L13R, W14R, and H929A single mutants, as well as AMBRA1[V10R/L13R/W14R] and AMBRA1[V10R/L13R/W14R/H929A] combined mutants, in promoting the ubiquitination of Cyclin D1-CDK4 (Fig. 3f, Supplementary Fig. 12a–b). V10R and L13R and combined mutations caused a complete loss of activity. Interestingly, W14R and H929A were less active than the wild-type protein despite having similar DDB1 binding as wild-type AMBRA1 (Supplementary Fig. 10b, Supplementary Fig. 12b). Our results imply that the N-terminal helix-loop-helix of AMBRA1 is essential for its association with the CRL4 complex, which in turn mediates the polyubiquitination of Cyclin D1. However, an efficient conjugation reaction appears to involve additional intermolecular interactions between DDB1 and both parts of the split WD40 domain.

## Mutations that disrupt the DDB1-AMBRA1 interface impair cell cycle progression

We further examined the effect of the interaction of DDB1 and AMBRA1 on the cell cycle in RB-proficient U2OS cells. Short interfering RNA (siRNA) oligoribonucleotides were used to knockdown endogenous AMBRA1 in cells (Fig. 4a, Supplementary Fig. 13). AMBRA1-silenced cells accumulated in S phase, accompanied by a decrease in the number of cells in G1 phase, compared to control cells (Fig. 4b). These results confirm previous reports that AMBRA1 coordinates cell cycle progression by regulating Cyclin D1 stability through the CRL4–DDB1 complex[9–11]. We then tested the effect of the re-expression of wild-type and mutated AMBRA1 in AMBRA1-silenced cells on cell cycle progression. Western blot analysis showed that all recombinant proteins were expressed in similar amounts (Fig. 4a, Supplementary Fig. 13). The percentages of V10R- and L13R-expressing cells in G1 significantly decreased compared to that of negative control and WT AMBRA1-re-expressing cells, while the percentage of S-phase cells was significantly increased. Similarly, W14R and H929A re-expressing cells accumulated in S phase relative to WT AMBRA1, but to a lower extent compared to V10R and L13R mutants, which were completely inactive in the Cyclin D1 ubiquitination assay (Fig. 4b, Supplementary Fig. 10b). Previously it was shown that AMBRA1 depletion results in an increase in replication

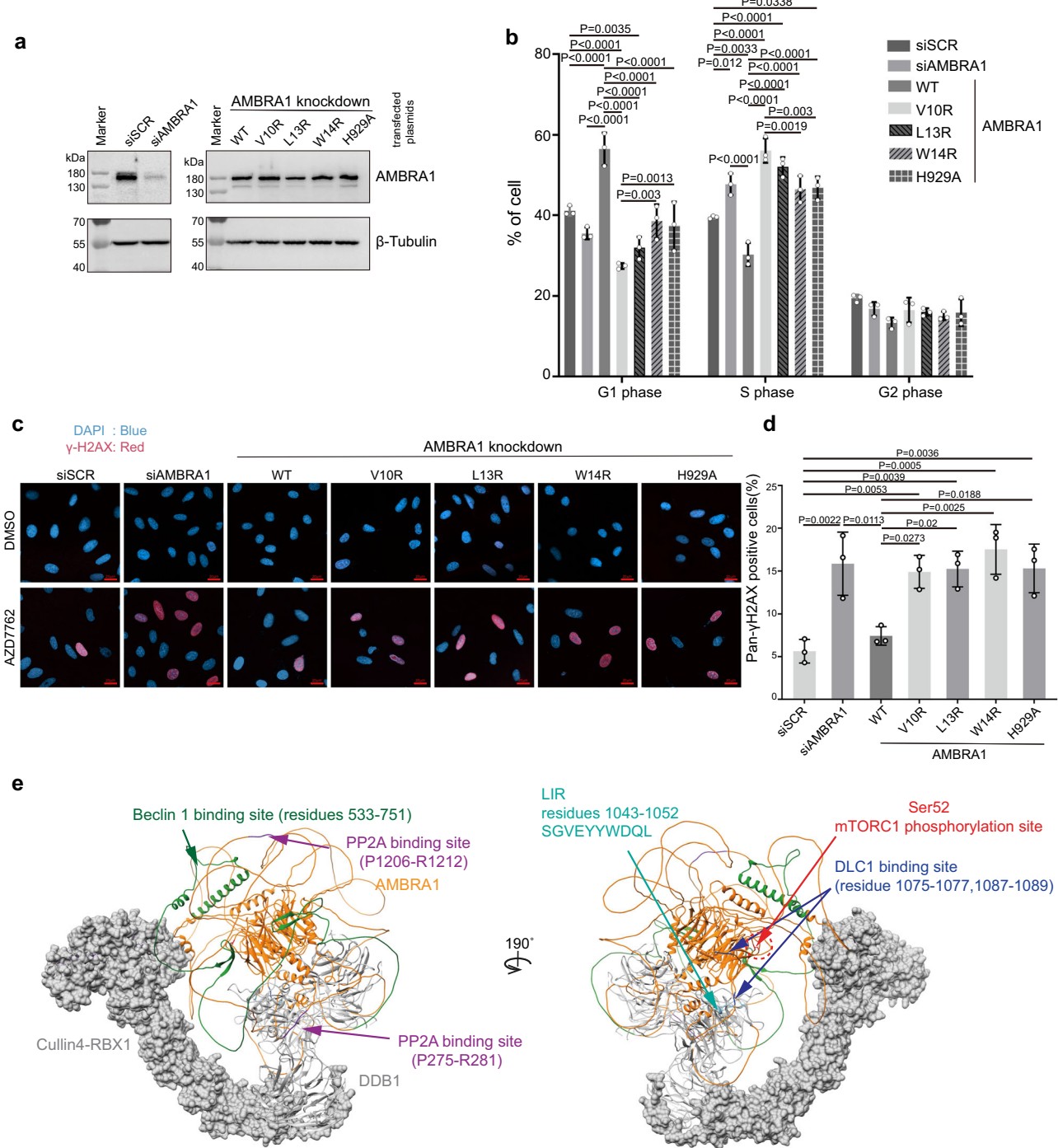

**e**

Beclin 1 binding site (residues 533-751)

PP2A binding site (P1206-R1212)

AMBRA1

PP2A binding site (P275-R281)

Cullin4-RBX1

DDB1

LIR residues 1043-1052 SGVEYYWDQL

Ser52 mTORC1 phosphorylation site

DLC1 binding site (residue 1075-1077, 1087-1089)

190°

stress and DNA damage, and this effect is greatly intensified by inhibition of CHK1[11]. We have investigated the effect of the re-expression of wild-type and mutated AMBRA1 on endogenous DNA damage in AMBRA1-silenced cells. Consistent with previous studies, we found that AMBRA1 downregulation results in a significantly increased DNA damage in cells treated with CHK1 inhibitor AZD7762 (Fig. 4c, d). Re-expression of wild-type AMBRA1 was sufficient to restore the non-interfered cell phenotype. None of the tested AMBRA1 point mutants was able to attenuate the high levels of endogenous DNA damage. In summary, V10 and L13 are the key binding sites, essential for AMBRA1 interaction with DDB1 and substrate ubiquitination, while W14R and H929A interface mutations only partially influence the ubiquitination activity and cause defects in downstream cellular functions. These results support a role for AMBRA1 in cell cycle regulation and demonstrate the importance of the AMBRA1$^{WD40}$-DDB1 interface in the CLR4−DDB1 complex.

## Discussion

This study provides insights into the structure and dynamics of AMBRA1, a central hub for the coordination of autophagy, the cell cycle, cell growth, development and apoptosis[5]. The highly disordered nature and "split" domain organization of AMBRA1 have largely precluded its structural determination. Intrinsically disordered regions may enable AMBRA1 to perform conformational changes to associate with different interactors simultaneously. To date, AMBRA1 has been reported to recruit and scaffold numerous substrates. For example, residues 533-751 are the binding site for Beclin1[12,27], residues 1075-1077 and 1087-1089 are required for dynein light chain 1 (DLC1)

**Fig. 4 | AMBRA1 regulates cell cycle progression and its deficiency sensitizes to CHK1 inhibition. a** AMBRA1 downregulation in U2OS cells using specific short interfering RNA (siRNA) oligo ribonucleotide. AMBRA1 level was analysed by western blotting, β-tubulin was used as the protein loading control. siSCR, scrambled siRNA. WT, wild-type. Source data are provided as a Source Data file. **b** Percentage of cells in the different phases of cell cycle after AMBRA1 knockdown, AMBRA1 WT and single mutant overexpression. U2OS cells were transfected with siSCR or siAMBRA1 using Lipofectamine RNAiMAX (ThermoFisher Scientific). Four days later, Cells were harvest for cell cycle analysis. Cell cycle analysis function of FlowJo 10.8.1 is used to derive the cell cycle populations. Data was then used to plot in GraphPad Prism 9.4.1 software. Statistical significance was determined by Tukey's multiple comparisons of two-way ANOVA, white round dots indicate individual data points for $n$ = 3 biological replicates. $P$ values are indicated in the graph. Graph bar was presented as mean values +/− SD. Source data are provided as a Source Data file. WT, wild-type. **c** Pan-γH2AX-positive cells in AMBRA1 knockdown, AMBRA1 WT and single mutant overexpression U2OS cells. Cells were treated with DMSO or 100 nM AZD7762 for 24 h, then harvested for immunofluorescence. Imaging was performed using a Zeiss LSM 900 confocal microscope with a Plan-Apochromat 40×/0.95 objective. DAPI was showed as blue colour, γH2AX was showed as red colour. Scale bar = 20 μm. **d** Quantification of Pan-γH2AX-positive cells in AMBRA1 knockdown, AMBRA1 WT and single mutant overexpression U2OS cells. For each group, total around 300 cells were manually counted by using ZEN (Blue edition) software, the percentage of pan-γH2AX positive cells were calculated. Data was then used to plot in GraphPad Prism 9.4.1 software. Statistical significance was determined by Tukey's multiple comparisons of one-way ANOVA, white round dots indicate individual data points for $n$ = 3 biological replicates. $P$ values are indicated in the graph. Graph bar was presented as mean values +/− SD. Source data are provided as a Source Data file. WT, wild-type. **e** Hypothetic model of Cullin4-RBX1-DDB1-AMBRA1 complex. Mapping the interactor binding sites on the Alphafold2 model of AMBRA1 superimposed on the AMBRA1[WD40]-DDB1 structure in this study and further aligned to Cullin4A-DDB1-RBX1 structures. Cullin4 and RBX1 are presented on the surface, and DDB1 and AMBRA1 are coloured in gray and orange respectively. The interactor binding sites are highlighted with different colours and labelled.

interaction[28], and residues S1043-L1052 are required for association with LC3 and induction of mitophagy[29]. Furthermore, the two PXP motifs (residues P275-R281 and P1206-R1212) were shown to interact with serine/threonine-protein phosphatase 2A (PP2A) and mediate dephosphorylation and degradation of the proto-oncogene c-Myc, resulting in inhibition of cell proliferation and tumorigenesis[30]. These reported regions are located at the IDRs linking WD40-N and WD40-C and the C-terminus. The N-terminal and C-terminal regions of AMBRA1 are involved in interactions with BCL-2, ULK1, ELONGIN B and PARKIN; however, the precise interaction sites are not known[31]. To gain insights into how these regions are organized relative to the AMBRA1 WD40 domain in the CRL4 structure, we superimposed the Alphafold2 model of AMBRA1, which retains all the loops, on our CRL4-AMBRA1 structural model (Fig. 4e). The AMBRA1 WD40 domain is a split domain that must reunite to form a functional substrate receptor. WD40 domain assembly and stabilization by DDB1 may function to organize and bring diverse substrates and CRL4 E3 ligase toward each other and thus assist in ubiquitin transfer, resulting in the ability of AMBRA1 to coordinate several biological processes in response to a variety of inputs. This mechanism is supplemented by the rotational flexibility of the DDB1 BPB domain relative to the rest of the DDB1-AMBRA1 complex. The DDB1 BPB domain is the attachment site for CUL4A and allows for long-range conformational changes bringing substrates and the E2 active site in close proximity. The IDRs contribute to a high degree of plasticity in AMBRA1 substrate recognition and recruitment. Several key interactions are mediated by the AMBRA1 WD40 domain itself. These include the mTORC1 regulatory phosphorylation site at Ser-52[6], as well as DDB1 and Cyclin D1 interaction sites. AMBRA1 regulates the stability of Cyclin D1 through interaction with DDB1 and the CLR4 complex. Cyclin D1 upregulation has been observed in several cancers, leading to hyperproliferation with genomic instability. We showed that the AMBRA1 WD40 domain alone is sufficient to mediate Cyclin D1/CDK4 interaction with DDB1 and Cyclin D1 in vitro ubiquitination. Residues V10 and L13 in the N-terminal helix-loop-helix motif are essential for the interaction between the AMBRA1[WD40] domain and DDB1. V10R and L13R mutations prevent interaction with DDB1, impair ubiquitination of Cyclin D1, deregulate cell cycle and result in the increased DNA damage. In addition to these key residues, W14 and H929 are also found at the interface between AMBRA1[WD40] and DDB1. W14 is the final hydrophobic residue in the first helix of the helix-loop-helix motif which anchors AMBRA1 to DDB1, while H929 provides an additional contact between WD40-C and DDB1. W14R and H929A mutations did not abolish the interaction with DDB1, but show similar functional defects. It is possible that W14R and H929A mutations influence the conformations of the AMBRA1[WD40]-DDB1 assembly, and may affect substrate binding or presentation for ubiquitination.

Since the AMBRA1 scaffold recruits a range of structurally diverse partners, one may expect high plasticity at these contact surfaces. WD40 domains can serve as hotspots for protein–protein interactions, and the flexibility of IDRs between sites where partners are bound may allow AMBRA1 to undergo conformational changes that promote interactions between diverse substrate proteins and the CRL4 complex. In this system, the AMBRA1[WD40]-DDB1 subcomplex serves as a base for substrate recruitment to the CLR4 active site. AMBRA1 plays a key role in tumorigenesis and progression, either as an oncogene or as a tumor suppressor[32]. This is consistent with a dual role of autophagy in cancer, either in suppressing the growth of tumors or in tumor growth promotion and chemoresistance[33–37]. The potential therapeutic strategy targeting AMBRA1 will therefore depend on the type of malignancy. Targeting AMBRA1 expression levels or regulatory interactions could increase the sensitivity of cancer cells to anticancer drugs or inhibit tumorigenesis and tumor progression. Thus, in providing structural information on AMBRA1, our work may ultimately lead to a better understanding of the molecular basis for the regulation of cell proliferation, autophagy and apoptosis and to the design of drugs targeting AMBRA1 functional interactions.

## Methods

### Antibodies

Anti-AMBRA1 antibody was used at 1:1000 dilution (CST, 24907s), anti-Phospho-Histone H2A.X (Ser139) antibody was used at 1:1000 dilution (CST, 2577s), Fab2 Fragment (Alexa Fluor 647 Conjugate) was used at 1:1000 dilution (CST, 4414s), anti-β-Tubulin Mouse Monoclonal antibody was used at 1:5000 dilution (Beyotime, AF2835), anti-c-Myc Mouse Monoclonal antibody was used at 1:2000 dilution (CWBIO, CW0299), HRP Conjugated Goat anti-Mouse IgG was used at 1:2000 dilution (CWBIO, CW0102), HRP Conjugated Goat anti-Rabbit IgG was used at 1:2000 dilution (CWBIO, CW0103).

### Cloning and mutagenesis

The gene for full length AMBRA1 (residue M1-R1298) was codon optimized. The genes encoding DDB1, Cullin4A, RBX1, Cyclin D1 and CDK4 were amplified from cDNA by PCR and subcloned into pCAG vectors with different tags using KpnI and XhoI cutting sites, respectively. For full length AMBRA1 used in in vitro experiments, it was constructed as a N-terminal MBP tag with a His6 tag in the C terminus. AMBRA1[WD40] (residues 1-204/853-1044) was subcloned with N-terminal GST tag followed by a TEV cutting site, or with N-terminal GST tag followed by a TEV cutting site and a Twin-Strep tag in the C terminus. For Cyclin D1, it was cloned with N-terminal GST-TEV-myc tag. DDB1, Cullin4A, RBX1 and CDK4 were constructed with N-terminal Twin-Strep-FLAG (TSF) tag. The AMBRA1 mutants were generated by PCR-based site-directed mutagenesis approaches. For AMBRA1[FL] (residue M1-R1298),

AMBRA1$^{N\Delta19}$ (residue A20-R1298), AMBRA1$^{N\Delta39}$ (residue M40-R1298), AMBRA1$^{V10R/L13R/W14R}$, AMBRA1$^{V10R/L13R/W14R/H929A}$ and AMBRA1 single mutants used in the MBP pull down experiment, they were cloned as a N-terminal MBP tag and a Twin-Strep tag in the C terminus. DDB1 was also subcloned into pFastBac-dual vector with a His$_6$ tag in the N terminus for insect cell expression. For in vivo cell experiments, taggless AMBRA1$^{FL}$, AMBRA1$^{N\Delta19}$, AMBRA1$^{N\Delta39}$, AMBRA1$^{V10R/L13R/W14R}$, AMBRA1$^{V10R/L13R/W14R/H929A}$ and AMBRA1 single mutants were cloned into pCAG vector. The primers used in this study are listed in Supplementary Table 3.

## Protein expression and purification

For mammalian cell expression, N-terminal GST tag human AMBRA1$^{WD40}$ and N-terminal TSF DDB1 were co-expressed for structural studies, N-terminal GST tag human AMBRA1$^{WD40}$ with C-terminal Twin-Strep tag was expressed for MS analysis, GST-Twin-Strep was expressed as negative control. N-terminal GST-TEV-myc tag human Cyclin D1 and N-terminal TSF tag CDK4 were co-expressed for in vitro ubiquitination experiment. Full length MBP tagged AMBRA1 with C-terminal His$_6$ tag was co-expressed with TSF tagged Cullin4A-DDB1-RBX1 E3 ligase for HDX measurement. MBP tagged full length AMBRA1, AMBRA1$^{N\Delta19}$, AMBRA1$^{N\Delta39}$, AMBRA1$^{V10R/L13R/W14R}$ and AMBRA1$^{V10R/L13R/W14R/H929A}$ with a C-terminal Twin-Strep tag were also expressed individually in Expi293F cells for in vitro pull-down experiment. N-terminal His$_6$ tag human DDB1 was expressed in sf9 cell.

The Expi293F cells were grown in Union-293 medium and used for protein expression, with 1 mg DNA and 4 mg PEI per 1 liter of cells at density of $2 \times 10^6$ cells/ml. Cells were harvested after 3 days and washed with 1× cold PBS.

For AMBRA1$^{WD40}$ and DDB1 co-expression, the cell pellet was lysed in lysis buffer (50 mM HEPES pH 7.4, 200 mM NaCl, 10% glycerol(v/v), 1% Triton-X100 (v/v), 2 mM MgCl$_2$, 5 mM beta-mercaptoethanol) with proteases inhibitors (1 mM PMSF, 0.15 μM Aprotinin, 10 μM leupeptin, 1 μM pepstain) for 20 min at 4 °C. After centrifugation at 39,191 × $g$ for 30 min, GST beads were incubated with the cell supernatant at 4 °C for 2 h. The beads were washed three times with wash buffer (50 mM HEPES pH 7.4, 200 mM NaCl, 2 mM MgCl$_2$, 5 mM beta-mercaptoethanol). Bound proteins were eluted with wash buffer containing 50 mM reduced glutathione. After TEV digestion overnight, the elution was then subjected to Streptactin sepharose resin, washed three times with wash buffer and eluted with gel filtration buffer (50 mM HEPES pH7.4, 200 mM NaCl, 2 mM MgCl$_2$ and 1 mM TCEP) supplemented with 5 mM desthiobiotin. The protein elution was further purified by Superdex 200 Increase 10/300 GL (GE Healthcare). Cyclin D1-CDK4 complex were purified in the same procedure as described above.

For full length AMBRA1 and Cullin4A-DDB1-RBX1 E3 ligase coexpression, the cell pellet was lysed in lysis buffer (50 mM HEPES pH 7.4, 200 mM NaCl, 10% glycerol(v/v), 1% Triton-X100 (v/v), 2 mM MgCl$_2$, 5 mM beta-mercaptoethanol) with proteases inhibitors (1 mM PMSF, 0.15 μM Aprotinin, 10 μM leupeptin, 1 μM pepstain) for 20 min at 4 °C. After centrifugation at 39,191 × $g$ for 30 min, amylose beads were incubated with the cell supernatant at 4 °C for 1 h. The beads were washed three times with wash buffer (50 mM HEPES pH 7.4, 200 mM NaCl, 2 mM MgCl$_2$, 5 mM beta-mercaptoethanol) and eluted in wash buffer containing 10 mM maltose. The elution was further applied to Streptactin sepharose resin, washed three times and eluted with gel filtration buffer containing 5 mM desthiobiotin. AMBRA1$^{N\Delta19}$, AMBRA1$^{N\Delta39}$, AMBRA1$^{V10R/L13R/W14R}$, AMBRA1$^{V10R/L13R/W14R/H929A}$ and AMBRA1 single mutants was purified in the same protocol.

The sf9 insect cell was grown in ESF921 medium and infected with recombinant baculovirus (1:50 v/v ratio) at density of $2 \times 10^6$ cells/ml. Cells were harvested after 3 days and washed with 1× PBS. For DDB1 purification, the cell pellet was lysed in lysis buffer (50 mM Tris pH 8.0, 200 mM NaCl, 20 mM imidazole, 1% Triton-X100 (v/v), 2 mM MgCl$_2$, 5 mM Beta-mercaptoethanol) with protease inhibitors for 20 min at 4 °C. After centrifugation at 39,191 × $g$ for 30 min, the supernatant was

passed over Ni-NTA affinity beads. The beads were washed three times with wash buffer (50 mM Tris pH8.0, 200 mM NaCl, 20 mM imidazole, 2 mM MgCl$_2$, 5 mM beta-mercaptoethanol) and eluted with elution buffer (50 mM Tris pH 8.0, 200 mM NaCl, 200 mM imidazole, 2 mM MgCl$_2$, 5 mM beta-mercaptoethanol). The protein elution was further purified by Superose 6 Increase 10/300 GL (GE Healthcare) in gel filtration buffer (50 mM HEPES pH 7.4, 200 mM NaCl, 2 mM MgCl$_2$, 1 mM TCEP). The protein sample was flash frozen in liquid nitrogen and stored at −80 °C until use. All purification procedures were carried out at 4 °C.

The proteins used in this study are shown in Supplementary Fig. 11.

## Baculovirus generation

For insect cell expression, target gene was cloned into pFastBac-dual vector, then transformed into DH10Bac Competent cell. The Bacmid was extracted and used to transfect the insect cell using Cellfectin II Reagent (Gibco) in 6-well plate, according to the manufacturer's instructions. 4 h after transfection, replace medium with fresh ESF921 medium containing 10% fetal bovine serum and 1% Antibiotic-Antimycotic solution. Cells were incubated at 27 °C for 4–5 days, then collect the supernatant and centrifuge at 1455 × $g$ for 5 min to remove cells and large debris, transfer the supernatant into new sterilized tube, that's P0 virus. To generate P1 virus, prepare 50 ml Sf9 cell with $2 \times 10^6$ cells/ml density, add 2 ml P0 virus into the cell, wait for 4–5 days and check the viability to drop to ~50–60%, Harvest the cell by centrifuge at 1455 × $g$ for 15 min, Collect the supernatant, that's P1 virus. Add FBS to a final concentration of 5% and store at 4 °C prevent from light. To generate P2 virus, follow the same procedure, the amount of P1 virus being used depending on the viral titers. For long-term storage, aliquot and store at −80 °C for later use.

## Hydrogen−deuterium exchange mass spectrometry (HDX)

H/D exchange was carried out by 10-fold dilution of 0.5 μM AMBRA1 or AMBRA1-CULLIN4-DDB1-RBX1 E3 ligase complex into a D$_2$O buffer (20 mM HEPES pH 7.5, 150 mM NaCl, 2 mM MgCl$_2$) to a final volume of 100 μL. After different time periods of incubation at 25 °C, the reaction was quenched by addition of ice-cold quench solution (1:1, v/v) containing 2 M guanidinium hydrochloride and 0.2 M citric acid. This reaction was digested on an immobilized pepsin column inside a manual HDX system with the temperature maintained at 0 °C. Eluted peptides were desalted using chilled trap column (1 mm × 15 mm, Acclaim PepMap300 C18, 5 μm, Thermo Fisher Scientific) for 5 min at a flowrate of 200 μL/min and 0.1% formic acid as mobile phase. Subsequent peptide separation was performed on the chilled ACQUITY BEH C18 (2.1 × 50 mm) analytical column using a first gradient ranging from 9 to 45 % of buffer B (80% acetonitrile and 0.1% formic acid) for 10 min followed by a second gradient ranging from 45 to 99% of buffer B for 1 min, at an overall flow rate of 50 μl/min. Peptides were ionized via electrospray ionization and analyzed by Orbitrap Eclipse and QExactive HFX (Thermo Fisher) mass spectrometer. Non-deuterated samples peptide identification was performed via data dependent tandem MS/MS experiments and analyzed by Proteome Discoverer 2.5 (Thermo Fisher). Mass analysis of the peptide centroids was carried by HDExaminer v3.3 (Sierra Analytics, Modesto, CA), followed by manual verification for each peptide. No corrections for back exchange that occurs during digestion and LC separation were applied. For peptides exhibiting a bimodal distribution each time point is fitted with two Gaussian peaks with different means and areas but with similar width into the spectra. Next, the fitted peak area parameter was used to calculate the relative amount of open state by taking the ratio of high and the sum of high and low mass subpopulation.

## Cryo-EM grid preparation and data acquisition

Two samples of human AMBRA1$^{WD40}$-DDB1 complex were prepared for cryo-EM data acquisition. The first one is the protein elution from strep

beads while the other one is the peak elution from size exclusion column. For cryo-EM grid, 3 μL of 0.15–0.2 mg/ml protein sample were deposited onto freshly glow-discharged Quantifoil R1.2/1.3 Cu300 mesh grids and plunged into liquid ethane using a FEI Vitrobot Mark IV after blotting for 3 s with blot force 0 at 4 °C and 100% humidity. A total of 6147 movies were collected on a Titan Krios electron microscope operating at 300 kV equipped with Gantan K3 camera at a defocus of −1 μm to −1.8 μm in counting mode, corresponding to a pixel size of 0.85 Å. Automated image acquisition was performed using SerialEM with a $3 \times 3$ image shift pattern. Movies consists of 50 frames, with a total dose of 58.96 $e^-$/Å$^2$, with a total exposure time of 3 s and a dose rate of 14.2 $e^-$/pixel/sec. Imaging parameters for the dataset are summarized in Supplementary Table 2.

## Cryo-EM data processing

The first dataset included 3456 movies and the another one has 2691 movies. For each dataset, the movies were firstly aligned using MotionCor2 wrapper in Relion 3 to correct the specimen movement and then imported into cryoSPARC v2[38–40]. CTF fitting and estimation were performed by patch CTF estimation. After particle picking with blob picker in the second dataset, 921,715 particles were picked and subjected to 2D classification. The good classes were selected as templates for the entire datasets, which generating 2,082,649 particles. The particles were cleaned up with iterative 2D classification and subjected for ab initio and non-uniform refinement, resulted in 3.0 Å resolution map. 780 K particles were imported into Relion 3.1 for 3D classification without alignment analysis. The map was postprocessed by deepEMhancer with tight target modes for model building[41]. All reported resolutions are based on the gold-standard FSC 0.143 criterion.

## Atomic model building and refinement

The coordinates for DDB1 (PDB: 2B5M) and AMBRA1 (identifier: AF-Q9C0C7-F1) downloaded from Alphafold2 Protein Structure Database were rigid body fitted separately into the density map using UCSF Chimera. Atomic coordinates were refined by iteratively performing Phenix real-space refinement and manual inspection and correction of the refined coordinates in Coot. To avoid overfitting, the map weight was set to 2 and secondary structure restraints were applied during automated real-space refinement. Model quality was assessed using MolProbity and the map-vs-model FSC by comparing the map-vs-model FSC with the FSC of the experimental cryo-EM density (Supplementary Fig. 7a). In order to evaluate overfitting, a cross-validation test was carried out. The atoms of the final coordinate model were displaced by an average of 0.5 Å, followed by one round of real-space refinement against one half map. The resulting map-vs-model FSC curves of the refined model against the same half map (work) and another half map (test) do not diverge significantly, indicating there is not overfitting (Supplementary Fig. 7b)[42]. Figures were prepared using UCSF Chimera version 1.15[43]. The cryo-EM density map has been deposited in the Electron Microscopy Data Bank under accession code EMD-37752 and the coordinates have been deposited in the Protein Data Bank under accession number 8WQR.

## In vitro ubiquitination assay

The ubiquitination assays were performed in a 25 μL reaction volume with the following components: 100 nM UBE1 (Boston Biochem E-304), 1.5 μM UBCH5C (Boston Biochem E2-627), 0.3 μM purified AMBRA1-DDB1-Cullin4-RBX1 complex, 20 μM HA-ubiquitin (Boston Biochem U-110), 0.16 μM myc-Cyclin D1-CDK4 complex and 10 mM MgATP solution (Boston Biochem B-20) in E3 ligase reaction buffer (Boston Biochem B-71). The reaction was incubated at 37 °C for 2 h and analysed by SDS-PAGE, followed by immunoblot. The experiment was repeated three times with similar result (Supplementary Fig. 12).

## RNA interference

SiRNA oligoribonucleotides corresponding to the human AMBRA1 were ordered from Tsingke. AMBRA1 siRNA 1: 5′- GAGUAGAACUG CCGGAUAG−3′; AMBRA1 siRNA 2: 5′-CCACCCAUGUGAACCAUAA-3′; AMBRA1 siRNA 3: 5′-GCGGAGACAUGUCAGUAUC-3′; AMBRA1 siRNA 4: 5′-CUGAAUCGCUGUCGUGCUU-3′; For siRNA transfection, a total of $8 \times 10^4$ cells per well were plated at 12-well plate. Each well transfected with optimal amount of siRNA mixture by using Lipofectamine RNAiMAX (ThermoFisher Scientific) next day, according to the manufacturer's instructions.

## Immunoblotting and immunofluorescence

The human osteosarcoma U2OS cells were plated into 12-well plate, then followed by siRNA transfection using Lipofectamine RNAiMAX (ThermoFisher Scientific) next day, according to the manufacturer's instructions. 24 h after siRNA transfection, transient plasmids transfections were performed using X-tremeGENE HP DNA Transfection Reagent (Roche) according to the manufacturer's instructions. Cells were harvested 48 h after plasmids transfection. Cells were lysed in lysis buffer (50 mM Tris pH 7.4, 150 mM NaCl, 0.5 mM EDTA, 1% NP-40, 0.1% SDS) plus protease inhibitors (1 mM PMSF, 0.15 μM Aprotinin, 10 μM leupeptin, 1 μM pepstain). Lysates were lysed for 30 min on ice, then centrifuged at $13,000 \times g$ for 10 min to remove insoluble debris. Solubilized proteins were quantified by BCA Protein Assay Kit (Sangon Biotech), equal amounts of protein were mixed with loading buffer and incubated at 95 °C for 5 min. whole cell lysates were separated by SDS-PAGE and transferred to 0.45 μm Immuno-Blot PVDF membranes (BIO-RAD). Membranes were then blocked in 5% milk/TBST for 1 h at room temperature and incubated with primary antibodies at 4 °C overnight. For the detection of proteins, using the appropriate secondary antibodies conjugated to horseradish peroxidase (anti-mouse and anti-rabbit, Cwbio) at 1:2000 dilution in TBST for 1 h at room temperature and visualizing with BeyoECL Plus (Beyotime). The images were acquired by Amersham Imager 680 (Supplementary Fig. 13).

For immunofluorescence, the human osteosarcoma U2OS cells were plated into the 12-well plate on coverslips at $5 \times 10^4$ density per well, then follow the same procedure as for immunoblotting. For CHK1 inhibition, cells were treated with 100 nM of AZD7762 (MedChemExpress) for 24 h after plasmids transfection, cells were washed with cold PBS and fixed in 4% Paraformaldehyde (PFA) in PBS for 20 min at room temperature, then permeabilized with PBS/0.1% (v/v) Triton X-100 for 20 min and subjected to blocking with PBS/1% (w/v) BSA to block the cell for 1 h at room temperature. γH2AX antibody (CST) were diluted in PBS/0.01% (v/v) Triton X-100 at 1:1000 dilution, applied onto the coverslips at 4 °C overnight. Alexa647-conjugated secondary antibody was diluted in PBS/1% (w/v) BSA at 1:1000 dilution and incubated with the coverslips for 1 h at room temperature. Slides were mounted in Anti-Fade Mounting Medium with DAPI (Beyotime). Imaging was performed using a Zeiss LSM 900 confocal microscope with a Plan-Apochromat 40×/0.95 objective.

## GST pull down assay

The N-terminal GST tagged AMBRA1$^{WD40}$ was co-expressed with a N-terminal TSF tag DDB1, the N-terminal GST-myc tag Cyclin D1 was co-expressed with a N-terminal TSF tag CDK4 in Expi293F cell. The complex was purified as described in the Protein expression and purification section. GST tag was remained for Cyclin D1-CDK4 complex. For GST pull down assay, 0.7 μM of AMBRA1$^{WD40}$-DDB1 complex was incubated with GST-myc-Cyclin D1-CDK4 complex in 5 to 1 molecular ratio supplemented with 50 μL GST beads for 1 h. After washing with 500 μL of gel filtration buffer three times, the proteins were eluted with 100 μL of gel filtration buffer supplemented with 50 mM reduced glutathione and resolved by SDS−PAGE. The experiment was repeated three times with similar result (Supplementary Fig. 14).

## MBP pull down assay

0.5 μM of purified AMBRA1$^{FL}$, AMBRA1$^{NΔ19}$, AMBRA1$^{NΔ39}$, AMBRA1$^{V10R/L13R/W14R}$ and AMBRA1$^{V10R/L13R/W14R/H929A}$ and four AMBRA1 single mutants (V10R, L13R, W14R, H929A) was incubated with 2.5 μM of purified DDB1 in 500 μL reaction volume supplemented with 50 μL amylose beads respectively. After 2 h incubation at 4 °C, the beads were washed three times with gel filtration buffer and eluted with 10 mM maltose containing buffer. The sample were further resolved by SDS–PAGE. The experiment was repeated three times with similar result (Supplementary Figs. 14 and 15).

## Cell cycle analysis

The human osteosarcoma U2OS cells were plated into 12-well plate at $8 \times 10^4$ density per well, then followed by siRNA transfection using Lipofectamine RNAiMAX (ThermoFisher Scientific) next day, according to the manufacturer's instructions. 48 h after siRNA transfection, Transient plasmids transfections of Ambra1 WT and single mutants were performed using X-tremeGENE HP DNA Transfection Reagent (Roche) according to the manufacturer's instructions. Cells were harvested 48 hrs after transfection. Using 0.25% Trypsin-EDTA (ThermoFisher Scientific) to detach the cell, collect the cell in 1.5 mL centrifuge tube by centrifuging at $1000 \times g$ for 5 min. Discard the supernatant carefully, then adds ice cold PBS to re-suspend the cell. centrifuge at $1000 \times g$ for 5 min. Remove the supernatant again and leave about 50 μL PBS. Tap the tube to avoid the clustering of the cell. Add 70% ice cold ethanol into the cell, gently pipette it upside down, leave in 4 °C to fix for 24 h. Centrifuge at $1000 \times g$ for 5 min to discard the supernatant carefully on the next day, then adds ice cold PBS to re-suspend the cell. Centrifuge at $1000 \times g$ for 5 min to collect the cell. Cells were then re-suspended with PI staining buffer containing RNaseA (Beyotime), incubate at 37 °C water bath for 30 min, prevent from the light. Then analysis the cell immediately by CytoFLEX S (Beckman). Data was analyzed by FlowJo 10.8.1.

Each treatment has three replicates. For data analysis, around $2 \times 10^4$ cells were recorded for each condition. All cells were first gated by SSC area vs. FSC area, and single cells were gated by FSC width vs FSC height, followed by PI height vs PI area. G1 cells were PI positive with 2 N DNA content, and G2/M cells were PI positive with 4 N DNA content, S cells were PI positive between 2 N DNA content and 4 N DNA content. Cell cycle analysis function of FlowJo 10.8.1 is used to derive the cell cycle populations. Then the total cell cycle population of G1, G2, S phases were normalized to 100%. Data was then used to plot in GraphPad Prism 9.4.1 software. Statistical significance was determined by Tukey's multiple comparisons of two-way ANOVA, $P$ values are indicated in the graph, the value of $p < 0.05$ was considered statistically significant. Graph bar was presented as mean values +/− SD.

## Quantification and statistical analyses

For Pan-γH2AX positive cells quantification, a minimum of three independent experiments were included in the representing graphs. For each group, total around 300 cells were manually counted by using ZEN (Blue edition) software, pan-γH2AX positive cells percentage were calculated. Data were analysed using GraphPad Prism 9.4.1 software. Statistical significance was determined by Tukey's multiple comparisons of one-way ANOVA, $P$ values are indicated in the graph, the value of $p < 0.05$ was considered statistically significant. Graph bar was presented as mean values +/− SD.

## Reporting summary

Further information on research design is available in the Nature Portfolio Reporting Summary linked to this article.

## Data availability

The atomic coordinates of the AMRBA1$^{WD40}$-DDB1 generated in this study and associated cryo-EM reconstruction have been deposited in the Protein Data Bank and EM data bank with the accession codes 8WQR and EMD-37752, respectively. Source data are provided with this paper. Previously solved structures used in this study were obtained from the PDB with accession codes: 6ZUE, 2B5L, 6FCV, 2HYE, 2B5M. The mass spectrometry proteomics data have been deposited to the ProteomeXchange Consortium via the PRIDE partner repository with the dataset identifier PXD046056[44].

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

## Acknowledgements

The authors thank the cryo-EM center (KEMC) and advanced mass spectrometry facility (KMS) of Kobilka Institute of Innovative Drug Discovery, the Chinese University of Hong Kong (Shenzhen) for the support. This work was supported by the Natural Science Foundation of Guangdong Province of China (to M.-Y.S., 2022A1515010856), the National Natural Science Foundation of China (to G.S., 31950410540), Foreign Youth Talent Program from State Administration of Foreign Experts Affairs (to G.S., QN2021032004L), Shenzhen-Hong Kong Cooperation Zone for Technology and Innovation (to G.S., HZQB-KCZYB-2020056), and the Start-up funding from SUSTech (to M.-Y.S.). M.-Y.S. is an investigator of SUSTech Institute for Biological Electron Microscopy.

## Author contributions

M.L., M.-Y.S. and G.S. designed the experiments. M.L. performed the experiments. Y.W. collected the EM data. F.T., X.W. and X.M. contributed in the early stages of the project. M.-Y.S. and G.S. processed the EM dataset. M.-Y.S. and G.S. wrote the manuscript.

## Competing interests

The authors declare no competing interests.
