## [Peer Review File · Nature Communications]

Structure of the DDB1-AMBRA1 E3 ligase receptor complex linked to cell cycle regulationREVIEWER COMMENTS

Reviewer #1 (Remarks to the Author):

Dear Liu et al.,

Presented here is a structure and dynamics characterization of the E3 ligase receptor AMBRA1 which is an integral member of ubiquitination, autophagy, and tumor suppression systems. You have provided valuable insight into how AMBRA1 complexes with DDB1 as well as HDX-MS analysis to support the theory of intrinsic disordered regions within the AMBRA1 protein. The pull down and activity assays provides a robust biological look into the impact of both the intrinsically disordered and helix-loop-helix regions of the protein.

Overall, the AMBRA1 complex analysis here presents important structural information that provides better mechanistic insight into critical biological processes. However, I feel that some of the conclusions made are not fully supported by sufficient information. I believe there are several elements that should be addressed to strengthen the manuscript for publication:

Major points

1. I think clarity is needed in the language used within, and in reference to, the HDX-MS section of the manuscript. HDX results demonstrate a protein's dynamics, but they do not alone provide information on domain organization or structural determination. While it does seem likely that there are IDRs in the AMBRA1 protein, I think that concluding that the presented HDX data proves intrinsic disorder is a faulty claim. There is a distinct difference between highly dynamic and intrinsically disordered. In the referenced book on HDX by David Weis (reference 15) there is ample discussion on characteristics of folded and unfolded structures – most of which are focused on the kinetics of the deuterium exchange (i.e. “fully solvated amides on the surface of structured segments of the protein also exchange readily albeit 1–2 orders of magnitude slower than those in an unstructured environment”). I would highly recommend the authors provide a more in-depth kinetics analysis of their HDX-MS data, similar to that shown for a single structured peptide in Fig. S2, to support the claim of IDRs in AMBRA1. In general, I think the AlphaFold structure is the best structural evidence in the paper that supports the claim of intrinsically disordered regions. Mapping deuteration level to the AF2 predicted structure may potentially further support the IDRs claim. Maybe consider moving the AlphaFold information to earlier in the manuscript instead of relying solely on HDX to characterize which regions are intrinsically disordered.
1. The HDX analysis could further be improved by making a more detailed / direct comparison between the HDX performed on AMBRA1 both in the apo state and with the E3 complex. Such a comparison would more robustly support the claims at the end of the HDX section of the “major stabilization effect” upon binding the CRL4 complex which is currently only shown for 5 selected peptides. This could potentially be easily visualized by adding the bound HDX data to the heatmap in Fig. S1.
2. The manuscript would benefit from statistical information on how “significantly reduced deuterium uptake” was calculated for IDR determination.
3. I would be interested to hear your interpretation for the low percent deuteration peptides within the IDRs (i.e. the peptide at residues 566-576). What about peptides in the unstructured region that show changes in deuteration over time (i.e. residues 692-698)? Would you infer this implies an intramolecular connection of the IDR loop to another part of the protein? Can this be mapped to the AlphaFold model?
4. Information is missing describing where the AlphaFold predicted models are from (is it just pulled from the AlphaFold Protein Structure Database: AF-Q9C0C7-F1 ?) and what the confidence levels of

the model are. Could AlphaFold be used to model the AMBRA1WD40 variant to compare to the cryo EM structure? I think a brief comment (1-2 sentences) on the pros and cons of using structure predictions compared to solving the structure experimentally, especially considering the disordered characteristic of the AMBRA1 protein, would be beneficial to the readers considering the significant rise in AlphaFold predictions being used in recent years.

Minor points

1. Overall, I think the paper could benefit from a crude diagram of the complex to demonstrate binding partners – this could easily be added to Fig 1.
2. The experimental methods and evaluation for the HDX-MS data are fairly standard and largely seem to follow recommendation for minimizing back exchange (Walters 2012). HDX is very pH dependent and notating the pH of the quench buffer in the Materials section would provide more information on its presumed effectiveness and subsequent back exchange. Is there a reason back exchange was not calculated? Or why experiments were not brought to full deuteration (via longer timepoints or increased pH)?
3. On page 4 line 130-131, “Secondary structure and domain prediction ...” references Fig. S1 of the deuterium uptake map. I think this sentence would be better supported by referencing the AlphaFold model of Fig. 5. Once the model depicting the structured domains is established, it would make sense to point out the black bars notated in Fig. S1.
4. Were there any efforts to normalize the deuteration levels both between timepoints and between the apo and E3 bound AMBRA1 HDX experiments (i.e. Fig. 1c, Fig. S1)?
5. Figure 1d is missing a label on the bottom row – presumably undeuterated apo AMBRA1?
6. When AMBRA1WD40 was coexpressed with DDB1 for cryo-EM, how was the stoichiometry determined? Also, was a mass standard run on the gel filtration chromatography system to confirm the elution profile agree with the additive mass of the complexed proteins?
7. On the top of page 6 line 170, the claim that the AMBRA1WD40 variant “does not disrupt the overall structure... of the protein” is a bit misleading. The protein is significantly truncated so the structure is inevitably changed (even if the change is just lost of IDRs). I would adjust this to say the structure of the WD40 domains is not disrupted, though be prepared to back this up.
8. The two point mutant variants AMBRA1V10R/L13R/W14R and AMBRA1V10R/L13R/W14R/H929A sufficiently prove that the helix-loop-helix region is essential for complex formation and biological function. It would be interesting to see how much individual residues effect the activity of the protein (i.e. is the V10R mutation alone enough to abolish or reduce activity on its own?). Is there a way that you could pinpoint which of the interacting residues contribute more or less to complex formation?
9. I think using the SDS-PAGE band density in Fig. 4 is a great way to quantify expression levels. It could be interesting to use this technique on other gels used in the paper (i.e. to quantify the reduction in ubiquitination activity of AMBRA1WD40 compared to the wildtype in Fig. 3d).
10. Consistent labeling of the lanes in the SDS-PAGE figures (i.e. add lane labels to Fig. 1b, point out the ubiquitinated D1 bands in Fig. 3, explain unlabeled bands such as in Fig. S7c).
11. I would personally suggest increased accessibility of data both by providing referenced anecdotal experiments (i.e. triplicate gels) in the supplemental information and by publishing the raw mass spectrometry data (in a repository) for access by readers.
12. A higher resolution/reformatted version of figure S1 would be beneficial as the text is quite small and difficult to read in its current state.

Overall, the work presented has potential to integrate new and diverse methods, AlphaFold prediction and HDX-MS, to both the structural biology and wider scientific audiences. The concept of truncating the AMBRA1 protein by removing supposed IDRs to solve parts of its structure by cryoEM will also significantly contribute to the understanding the ubiquitin conjugation system and its complexes. However, in its current state, I believe there are several additional data analyses that are needed to sufficiently support the claims made in the manuscript.

Reference:

1. Walters, B.T. et al. (2012) "Minimizing back exchange in the hydrogen exchange-mass spectrometry experiment," *Journal of the American Society for Mass Spectrometry*, 23(12), pp. 2132–2139. <https://doi.org/10.1007/s13361-012-0476-x>.

Reviewer #2 (Remarks to the Author):

The primary purpose of your review is to provide feedback on the soundness of the research reported. This will help authors to improve their manuscript and editors to reach a decision. When composing your report, the following questions might assist you in writing a well-justified review, but please feel free to raise any further questions and concerns about the paper.

- What are the noteworthy results?
- Will the work be of significance to the field and related fields? How does it compare to the established literature? If the work is not original, please provide relevant references.
- Does the work support the conclusions and claims, or is additional evidence needed?
- Are there any flaws in the data analysis, interpretation and conclusions? Do these prohibit publication or require revision?
- Is the methodology sound? Does the work meet the expected standards in your field?
- Is there enough detail provided in the methods for the work to be reproduced?

Where applicable, reporting summaries are requested from the authors to improve the transparency and reproducibility of published results. We hope the file, if included, will aid in your evaluation of the paper as they contain key information pertaining to study design and analysis.

To increase the transparency and openness of the reviewing process, we support our reviewers signing their reports to authors if they feel comfortable doing so. If, however, you prefer to send an anonymised report we will continue to respect and maintain your anonymity. The comments to the authors are subsequently shared with the other reviewers, but your identity will not be revealed unless you have signed your report.

Key results:

The authors isolate and structurally characterize an engineered mutant of AMBRA1, a tumor suppressor, in complex with DDB1, an adaptor that is part of the ubiquitination pathway. The portion of AMBRA1 used in this study includes an N-terminal helix-loop-helix motif which the authors show to be essential for binding to DDB1, and a non-canonical WD40 domain comprised of both an N terminal and a C terminal segment of AMBRA1, with the long intrinsically disordered region between those segments deleted. The authors fit an existing DDB1 model and an AlphaFold model prediction of the AMBRA1 WD40 domain into their 3 Å EM density map and refined the coordinates. Their EM/AlphaFold modelling observations are consistent with their HDMX observations. The authors conclude that, when the residues observed to be in the binding interface of their protein model were mutated, the mutations resulted in loss of function in in vitro (pull-down, ubiquitination) and in vivo (immunofluorescence) experiments.

Overall Feedback:

Because of the role of AMBRA1 in multiple pathways including cancer, this work is likely to be important for the field. However, the introduction and parts of the discussion are difficult to follow. There are many components to this system. In the introduction, the authors introduce about 15 different components. For scientists outside this field, it is unclear how these components interact and what roles they play. Perhaps it would be helpful to include a schematic or cartoon representation of

each of the parts and how they interact / what role they play. The introduction may need to be expanded to more clearly define the components and what is important for the reader to know about them in order to understand the main findings of this study. At minimum, the introduction could be broken up into distinct paragraphs so that the reader has a clear indication of when a different role of AMBRA1 is being described. On a related note, there are times the authors refer to a substrate and it is not clear to me what substrate the reader is meant to understand.

Additional data are needed to fully characterize the EM results presented. The authors state that their reconstruction is "around 3 Å" resolution, but also describe that one portion of the map (BPB domain, line 180-182) is quite flexible. It would help the reader if the authors include a map of local resolution to more clearly display the quality of the reconstruction which, as with most EM reconstructions, varies across the map. Additionally, the map appears to display some anisotropy, suggesting a preferred orientation of particles. It would be helpful for the authors to also include an orientation distribution plot so the reader can understand whether or to what degree a preferred orientation was observed.

There are some grammar choices that negatively impact the clarity of the writing. I am noting some of those which are most confusing. There are other instances of errors in subject-verb agreement and use of plural or singular nouns throughout the manuscript which are not captured here.

Specific points

- Line 37-38, What is meant by "highly" intrinsic disorder?
- Line 46 – 48 – the italicized portion is unclear "AMBRA1 uses its N-terminal helix-loop-helix and WD40 domain to bind the double-propeller fold of DDB1, whereas different regions target the specific cellular substrates for ubiquitination." Different regions of what? What are the specific cellular substrates?
- Line 51 – subject-verb agreement error – "these results...suggest"
- Line 74 – There is not a clear connection between the first and second sentence.
- Lines 87 – Which protein is "its" referring to? "In nutrient-rich status, mTORC1 inhibits AMBRA1 by phosphorylation on its Ser-52 [6, 7]."
- Line 89 – What is meant by "In addition to ULK1..."? This sentence structure is confusing. If "AMBRA1 functions as a substrate receptor for Beclin1 ubiquitination," what is the substrate in this reaction? A substrate of what enzyme? This is an example where I think a schematic figure may help.
- Lines 90 – 93 mention Beclin1 multiple times without defining what it is. Please also define PI3KC3 (line 92), CDK4/6 (line 96), and D-type cyclins (lines 94, 95). This section is an example of where it is unclear to the reader how these different components of the system interact.
- Line 106 – "suggests a mechanism by which the AMBRA1 targets proteins involved in essential biological processes" Might the authors narrow this statement? This work goes on to compare structures of other other target proteins, but all within the same family, with very similar structures.
- Line 114 – "AMBRA1 has been characterized by WD40 domain in the N-terminal region while the rest of the protein is predicted as intrinsically disordered regions (IDR) (Fig. 1a) [12]." This wording is unusual and unclear. If a protein has been characterized by X, I would expect X to be a technique such as cryoEM or crystallography or even by functional studies. Perhaps AMBRA1 is predicted to have a WD40 domain, in which case I would still ask what this prediction is based on. In my effort to understand what was meant by this statement, I see that this language is very close to that used in reference 12, which is a review that cites for this statement, another review. Has a WD40 domain been observed in this protein? A homologue? Many homologues?
- Fig.1 –
 - o Panel b, what is the middle lane? Please label the lanes.
 - o Panel c figure legend – what is meant by the "calculated central residue of an individual peptide?"

o Panel d – What is the bottom row? I think it makes sense to swap the top and middle rows so the reader sees the AMBRA1 alone data first, and then AMBRA1 + E3 Ligase. This would be more consistent with how the figure legend and the main body text are written.

- Fig. S1 - I cannot read any of the grey text in Fig. S1. It is too small.

- Fig. S2 – What is the top row of mass spectra? Is this time zero? Please label. Would it make sense to use panels here for the bimodal isotopic envelopes (panel a) vs the kinetics (panel b), vs the structure (panel c)? In the structure, please define the colors used (red, blue, grey).

- Results lines 110-160. Do the open/closed states observed by HDX-MS correspond to folded/unfolded structures?

- Figure 2 –

- o Panel C – The coloring is different in the model shown in the EM map as opposed to the model depicted on it's own. The two shades of green and the two shades of blue are very difficult to distinguish.

- o Panel D –

Please describe why these sections are highlighted. Does the left panel show the hydrophobic interactions and the right show the ionic interactions?

The text (lines 208-209) identifies highlighted residues from BPC including A381. The figure legend identifies BPC as being colored in blue. The left panel shows A381 but it is colored green. The text and figure appear inconsistent with each other.

- Figure 3 –

- o Panel b – In general, these bands are quite faint. Could the contrast be increased? Alternatively perhaps run the gels with more sample loaded. Why was the delta19 variant included?

- o Panel c – why are there bands observed in the pull down that are not observed in the input?

- o Panels d, e, and f – Please label all the observed bands. It is not clear, in panel e in particular, where there are multiple bands present in conditions the authors explain have no ubiquitination activity.

- Line 175 – As noted above, more detail is needed with respect to global and local resolution as well as preferred orientation.

- Fig. S3.

- o Panel c – please include how many particles were included at each step and resolution at each step.

- o The GSFSC and the portion of the EM map shown at the lower right need panel labels and descriptions in the figure legend.

- Fig. S4.

- o Panel a – I'm used to seeing a map vs model as one curve, not two separate curves. If we are to compare these two curves, they look pretty different.

- o Panel c – The smaller portions appear to be chosen to show side chain densities, and the larger portions to show whole domains. The larger domain views are too busy to make out details. Consider showing where in the domain structures the smaller highlighted portions are taken from.

- Fig. S5. Please describe the figure in the legend. Are these different classes in 3D classification?

Something else? How many particles are present in each class? 3D classification is mentioned in the main text, but it is not shown as part of the EM processing workflow (Fig. S3)

- Lines 206 – 209: The N-terminal helical motif is almost entirely engulfed by the large pocket formed between BPA and BPC double-propeller of DDB1, stabilizing between the hydrophobic interactions contributed by W953, F1003, V360, L814, A381 of BPC domain with V10, L13, W14 in the first helix.

- o “stabilizing between” is confusing. Perhaps the authors mean stabilized by?

- o At the end of the sentence, “in the first helix” seems incomplete. Please state which protein is meant by this clause, possibly “in the first helix of AMBRA1.”

- Lines 213 – 215: We incubated purified MBP tagged AMBRA1FL, AMBRA1NΔ19 or

AMBRA1V10R/L13R/W14R and AMBRA1V10R/L13R/W14R/H929A with His6 tagged DDB1.

- o The “and” is confusing. Seems to imply that AMBRA1V10R/L13R/W14R/H929A was included in all experiments with one of the other variants.

- Lines 220 – 223 has grammar errors that make it difficult to understand.

o As written: Comparison of our AMBRA1-DDB1 structure to other DDB1 complex, the binding between DDB1 and AMBRA1WD40 is similar to other adaptors bound to DDB1 including DCAF1 (PDB:6ZUE), Simian virus 5V (PDB: 2B5L), CSA (PDB: 6FCV) , DDB2 (PDB: 3EI4), suggesting a common molecular mechanism of DDB1 recognition (Fig. S6b).

o Potential revision: Comparison of our AMBRA1-DDB1 structure to other DDB1 complexes shows that the binding between DDB1 and AMBRA1WD40 is similar to other adaptors bound to DDB1 including DCAF1(PDB:6ZUE), Simian virus 5V (PDB: 2B5L), CSA (PDB: 6FCV) , DDB2 (PDB: 3EI4), suggesting a common molecular mechanism of DDB1 recognition (Fig. S6b).

- Fig. S6

o Panel B – the referenced structures are all of an adaptor-DDB1 complex. The figure overlays the structures of the different adaptors. It appears there is only one model displayed for the DDB1. Is the DDB1 structure exactly the same in all of these complexes?

- Lines 226 – 232 has many grammar errors that are hard to understand. The below has possible revisions. This section also asks the reader to visualize movements within the proposed structural model but, again, there are many components involved. Could the authors present this in a schematic or even a movie?

o Our model revealed a ~70 Å distance between RBX1 and central cavity of AMBRA1. Notably, the 3D classification without alignment analysis confirmed the plasticity of BPB domain of DDB1, which can adopt diverse orientations to position AMBRA1 for substrate (what is the substrate?) ubiquitination. RBX1 is likely to reposition as a result of neddylation, therefore establishing an open active conformation to mediate transfer of ubiquitin from E2 to substrate, which shortens the distance between the substrates and the enzymes.

- Line 247: AMBRA1 functions as a substrate receptor to recruit Cyclin D1.

o What is Cyclin D1 recruited to? What is the substrate?

- Line 247 – 249: Therefore, disrupting the interaction between AMBRA1 and DDB1 would fail to mediate the ubiquitination of its substrate.

o This sentence is confusing. What is doing the mediation? What does “its” refer to?

- Lines 249 – 256 The authors mention poly-ubiquitination twice, note conditions in which they “did not observe any ubiquitination” and also mention “ubiquitination.” Is there an important distinction to be made between ubiquitination and polyubiquitination?

- Lines 252 – 256 We further tested the effect of AMBRA1 mutants including AMBRA1V10R/L13R/W14R and AMBRA1V10R/L13R/W14R/H929A in promoting the ubiquitination of Cyclin D1-CDK4 (Fig.3f), none of these mutants execute the activity, indicating the N terminal helix-loop-helix of AMBRA1 is essential for association with CRL4, which in turn mediate the polyubiquitination of Cyclin D1.

o The sentence makes a conclusion about the N terminal helix-loop-helix domain, but the H929A mutation is not in that domain.

- Line 263 – What is LC3-II and why is it important? How is LC3-11 involved in autophagy?

- Line 276-277 This was further supported by LC3 immunoblotting results showing that AMBRA1 can induce autophagy by interacting with CRL4 E3 ligase. (Fig.4d-e).

o E3 ligase is not included in the figure mentioned in this sentence. What is the connection?

- Fig. 5 – This figure needs a more complete description in the figure legend. There are grey portions shown in surface and in ribbon model, and multiple different colors used in the ribbon model. Is any experimental evidence from this paper used in this figure, or are all the structures shown AlphaFold2 predictions and the PDB 2HYE structure used in Figure S6?

- Fig. S7

o Panel b – what is the band ~70 kDa?

o Panel d – There are 2 bands. I can't tell if the label is one label or two labels. The bottom line of the label is missing the “-” which is present in the other labels on this figure indicating the position of the band corresponding to the label.

- Methods

- o Line 662. This portion begins with incubating the “cell supernatant” with amylose beads. There is no mention to the lysis and centrifugation steps that, in the previous paragraph, precede this step. Please include all steps of this purification.
- o Line 666 – What is meant by “Full length AMBRA1 N19?” How is the construct both full length and a truncation?

Reviewer #3 (Remarks to the Author):

Liu et al characterized the structure of the AMBRA1 beta-propeller, focusing on its interaction with DDB1, which is functional to Cyclin D1 ubiquitylation. The relevance of this pathway is highlighted by three papers recently published on *Nature*, showing the importance of the AMBRA1/DDB1/CyclinD1 axis in cancer and neurodevelopment.

However, although of potential interest, a series of findings in the article make it not entirely novel or sound. Moreover, the intriguing and potentially groundbreaking findings regarding AMBRA1 structure are not fully balanced by any complete associated functional characterizations of the underlying biological mechanisms.

Despite the claims of the authors in the abstract (line 42-43: “Here we present the cryo-EM structure of AMBRA1 in complex with DDB1 at 3 Å resolution”), the structure itself is far from being totally understood, since they describe only a portion of the protein whose function has been already characterized by others (PMID 25499913; PMID 33854232), who identified the region for binding with DDB1 in the first 180 residues and the relevance of this region for Cyclin D degradation. Also, the statement regarding the hypothesized WD40-N – WD40-C folding (lines 135-141) is misleading and should be removed or rephrased. Last, put in this manner, the reader may think that this is a totally novel concept, but the exact same conformation was already proposed and published this year (PMID 36243772, not cited in the manuscript), taking advantage of the AlphaFold2 prediction. Of note, the results here cited are coherent with the experimental data provided by the authors. The title as well is not fully appropriate. Despite in Fig. 3d the authors evaluate the ability of wildtype and mutant AMBRA1 to recruit and promote cyclin D ubiquitination, there is no structural data regarding AMBRA1 substrates (e.g., cyclin D) recruitment, since the manuscript is largely focused on the upstream AMBRA1-DDB1 interaction.

On a positive note, I found extremely intriguing the multiple AMBRA1WD40 single and multiple residues mutants. Are they mutated in cancer? Do they have any effect on cell cycle regulation? What about their sensitivity to CHK1 inhibitors?

Could the authors better define what are the physio-pathological consequences of this mutants apart from cyclin D1 ubiquitination?

Being the AMBRA1WD40 able to interact per se with its substrate cyclin D1 (as demonstrated in Fig. 3c), one may suppose that also other substrates of its scaffolding activity for CRL4 ubiquitylation could interact similarly. Consequently, it would be interesting to define its interactome to identify novel targets of its DCAF activity, while potentially excluding the plethora of other AMBRA1 interactors.

Additionally, the manuscript contains a series of flaws here listed:

- Liu et al provide a further structural characterization by narrowing down the interaction region to the first 49 residues plus some others present in the C-Term WD40 domains. This has been demonstrated

through cryoEM experiments, in vitro binding, and ubiquitylation assays carried with an NΔ49 mutant. However, the NΔ49 deletion would likely disrupt the local propeller structure, and it is not a proof of direct binding.

- The authors state that the residues H929 and D866 in the C-TERM WD40 could mediate ionic bonding with DDB1, but this is not experimentally validated with a sufficient level of detail. In Fig 3b the authors investigate the AMBRA1 V10R/L13R/W14R/H929A and AMBRA1V10R/L13R/W14R mutants binding with DDB1. However, the AMBRA1V10R/L13R/W14R mutations already completely abolish the interaction, making it difficult to determine the influence of the H929A substitution on the interaction. A single mutant H929A should be done to evaluate the impact of the residue for AMBRA1/DDB1 binding. Moreover, these multiple mutations on AMBRA1 could disrupt the local propeller structure rather than impacting the DDB1 interaction. This aspect should be carefully checked, also considering that in Fig. 3b the mutant proteins seem to be less expressed as a possible result of destabilizing mutations.

- The experiments suggest the importance of the N-TERM helix loop helix in AMBRA1 DDB1 interaction but do not entirely rule out or fully characterize the importance of other regions in the DDB1-AMBRA1 binding.

- In Fig. 4 the authors investigate the role of AMBRA1 depletion in autophagy through reconstitution experiments with a wild type or NΔ49 mutant. This part of the manuscript is definitely underdeveloped with respect to the previous sections and does not add anything novel enough. Indeed, the methods used in this study are not rigorous nor up-to-date, therefore it is impossible to draw any conclusion from the results obtained. Figures 4a-c need a proper autophagic flux assay, that typically include starvation/cloroquine or bafilomycin (as the authors do in Fig. 4d). In Fig. 4d the endogenous AMBRA1 signal is not visible so it is impossible to assess the efficiency of the AMBRA1 interference. Again in Fig. 4d, to prove the efficacy of Bafilomycin treatment unprocessed LC3 I signal is necessary.

Moreover, since the manuscript focuses on the structural characterization of AMBRA1 beta-propeller, DDB1 interaction, and Cyclin D1 ubiquitylation, the most correct in-cell validation should focus on the control of cell cycle progression and its consequence on genome instability.

REVIEWER COMMENTS

Reviewer #1 (Remarks to the Author):

Dear Liu et al.,

Presented here is a structure and dynamics characterization of the E3 ligase receptor AMBRA1 which is an integral member of ubiquitination, autophagy, and tumor suppression systems. You have provided valuable insight into how AMBRA1 complexes with DDB1 as well as HDX-MS analysis to support the theory of intrinsic disordered regions within the AMBRA1 protein. The pull down and activity assays provides a robust biological look into the impact of both the intrinsically disordered and helix-loop-helix regions of the protein.

Overall, the AMBRA1 complex analysis here presents important structural information that provides better mechanistic insight into critical biological processes. However, I feel that some of the conclusions made are not fully supported by sufficient information. I believe there are several elements that should be addressed to strengthen the manuscript for publication:

We very much appreciate the referee's comments and suggestions.

Major points

1. I think clarity is needed in the language used within, and in reference to, the HDX-MS section of the manuscript. HDX results demonstrate a protein's dynamics, but they do not alone provide information on domain organization or structural determination. While it does seem likely that there are IDRs in the AMBRA1 protein, I think that concluding that the presented HDX data proves intrinsic disorder is a faulty claim. There is a distinct difference between highly dynamic and intrinsically disordered. In the referenced book on HDX by David Weis (reference 15) there is ample discussion on characteristics of folded and unfolded structures – most of which are focused on the kinetics of the deuterium exchange (i.e. “fully solvated amides on the surface of structured segments of the protein also exchange readily albeit 1-2 orders of magnitude slower than those in an unstructured environment”). I would highly recommend the authors provide a more in-depth kinetics analysis of their HDX-MS data, similar to that shown for a single structured peptide in Fig. S2, to support the claim of IDRs in AMBRA1. In general, I think the AlphaFold structure is the best structural evidence in the paper that supports the claim of intrinsically disordered regions. Mapping deuteration level to the AF2 predicted structure may potentially further support the IDRs claim. Maybe consider moving the AlphaFold information to earlier in the manuscript instead of relying solely on HDX to characterize which regions are intrinsically disordered.

Thank you for this important observation. As suggested, we have now included more in-depth kinetic analysis of HDX data. Specifically, we plotted exchange curves for number of peptides mapping to WD40 and proposed IDR containing regions (now Fig. S4). Intrinsic disorder in AMBRA1 was proposed previously due to lack of recognizable protein domains. AlphaFold and other bioinformatic studies support this conclusion. Many peptides from these long IDR regions exchange rapidly within several seconds, while WD40 domain exhibits much slower HDX exchange. Some secondary structure may exist within IDR regions in form of alpha helices and beta harpins as predicted by AlphaFold – showing reduced deuteration. Additionally, molecular interactions may be accompanied by a disorder-to-order transition. We have now mapped HDX data to AlphaFold model and revised the manuscript to clarify these observations and implications of HDX results.

1. The HDX analysis could further be improved by making a more detailed / direct comparison between the HDX performed on AMBRA1 both in the apo state and with the

E3 complex. Such a comparison would more robustly support the claims at the end of the HDX section of the “major stabilization effect” upon binding the CRL4 complex which is currently only shown for 5 selected peptides. This could potentially be easily visualized by adding the bound HDX data to the heatmap in Fig. S1.

Thank you for this suggestion. Because AMBRA1 contains EX1 kinetics and multiple conformational states – it is challenging to show results as a simple difference heat map. The bound form is still conformationally heterogeneous. However, we have now included kinetics data for 30 AMBRA1 peptides representing WD40 and IDR regions in both apo and E3 bound state.

2. The manuscript would benefit from statistical information on how “significantly reduced deuterium uptake” was calculated for IDR determination.

AMBRA1 HDX experiments were conducted in triplicates and SD (0.1 #D) was calculated using HDXExaminer and reported in Supplementary table 1. No additional statistical analysis was performed to determine IDR. We revised the text and removed “significantly” to avoid confusion. From Figure 1 we can see that peptides from WD40 exchange gradually from 0 to about 50%, while most of the IDR peptides exchange to 70-80% within 10s and maintain this level over the course of deuteration. Please note that data were not corrected for back-exchange. In our HDX-system back-exchange is about 18% (observed from other projects).

3. I would be interested to hear your interpretation for the low percent deuteration peptides within the IDRs (i.e. the peptide at residues 566–576). What about peptides in the unstructured region that show changes in deuteration over time (i.e. residues 692–698)? Would you infer this implies an intramolecular connection of the IDR loop to another part of the protein? Can this be mapped to the AlphaFold model?

This is an excellent point. AlphaFold prediction shows presence of few secondary structure elements within IDR. It is important to note that these few helices and beta hairpins do not fold together into recognizable domain. Residues 566-576 and 692-698 map to these secondary structure elements. Interestingly Beclin1 binding site was previously mapped to region 533-751, and intramolecular interaction of these IDR regions and another part of the protein cannot be excluded. We have mapped to the AlphaFold model as suggested in Fig. S2, and included brief description in the text.

4. Information is missing describing where the AlphaFold predicted models are from (is it just pulled from the AlphaFold Protein Structure Database: AF-Q9C0C7-F1 ?) and what the confidence levels of the model are. Could AlphaFold be used to model the AMBRA1WD40 variant to compare to the cryo EM structure? I think a brief comment (1–2 sentences) on the pros and cons of using structure predictions compared to solving the structure experimentally, especially considering the disordered characteristic of the AMBRA1 protein, would be beneficial to the readers considering the significant rise in AlphaFold predictions being used in recent years.

The Alphafold model was downloaded from AlphaFold Protein Structure Database (AF-Q9C0C7-F1). We have now included the information in the manuscript. AlphaFold impact to structural biology is certainly enormous, but the experimental structures are needed to both verify predictions, determine side chain configuration and details of protein complexes, as well as conformational changes and dynamics. We feel that discussion about AlphaFold impact would be out of scope our manuscript.

Minor points

1. Overall, I think the paper could benefit from a crude diagram of the complex to demonstrate binding partners – this could easily be added to Fig 1.

This has been added to Fig 1a.

2. The experimental methods and evaluation for the HDX–MS data are fairly standard and largely seem to follow recommendation for minimizing back exchange (Walters 2012). HDX is very pH dependent and notating the pH of the quench buffer in the Materials section would provide more information on its presumed effectiveness and subsequent back exchange. Is there a reason back exchange was not calculated? Or why experiments were not brought to full deuteration (via longer timepoints or increased pH)?

AMBRA1 can be purified in only small amount and is highly unstable. Therefore, we did not carry out 100% control experiment. However, based on empirical experience from other projects – back-exchange on our system is about 18%.

3. On page 4 line 130–131, “Secondary structure and domain prediction ...” references Fig. S1 of the deuterium uptake map. I think this sentence would be better supported by referencing the AlphaFold model of Fig. 5. Once the model depicting the structured domains is established, it would make sense to point out the black bars notated in Fig. S1.

We have modified the sentence.

4. Were there any efforts to normalize the deuteration levels both between timepoints and between the apo and E3 bound AMBRA1 HDX experiments (i.e. Fig. 1c, Fig. S1)?

No, these were separate experiments.

5. Figure 1d is missing a label on the bottom row – presumably undeuterated apo AMBRA1?

This has been added.

6. When AMBRA1^{WD40} was coexpressed with DDB1 for cryo–EM, how was the stoichiometry determined? Also, was a mass standard run on the gel filtration chromatography system to confirm the elution profile agree with the additive mass of the complexed proteins?

We estimated the stoichiometry between AMBRA1^{WD40} and DDB1 was 1:1, which was determined by the elution position from gel filtration. We have run a calibration kits which contained a well-defined series of protein standards, then the molecular weight of AMBRA1^{WD40}-DDB1 complex was determined by comparing its elution volume with those of known protein standards.

7. On the top of page 6 line 170, the claim that the AMBRA1^{WD40} variant “does not disrupt the overall structure... of the protein” is a bit misleading. The protein is significantly truncated so the structure is inevitably changed (even if the change is just lost of IDRs). I would adjust this to say the structure of the WD40 domains is not disrupted, though be prepared to back this up.

Thanks for the suggestion. This has been modified.

8. The two point mutant variants AMBRA1V10R/L13R/W14R and AMBRA1V10R/L13R/W14R/H929A sufficiently prove that the helix–loop–helix region is

essential for complex formation and biological function. It would be interesting to see how much individual residues effect the activity of the protein (i.e. is the V10R mutation alone enough to abolish or reduce activity on its own?). Is there a way that you could pinpoint which of the interacting residues contribute more or less to complex formation?

We have prepared the AMBRA1 single mutants and tested them in DDB1 interactions, in vitro ubiquitination and in vivo cell experiments. The single mutation within the helix-loop-helix abolished the interaction with DDB1, and therefore could not efficiently mediate the ubiquitination of Cyclin D1.

9. I think using the SDS-PAGE band density in Fig. 4 is a great way to quantify expression levels. It could be interesting to use this technique on other gels used in the paper (i.e. to quantify the reduction in ubiquitination activity of AMBRA1WD40 compared to the wildtype in Fig. 3d).

We thank the suggestion from the reviewer. However, these bands from western blot are difficult to use for quantification.

10. Consistent labeling of the lanes in the SDS-PAGE figures (i.e. add lane labels to Fig. 1b, point out the ubiquitinated D1 bands in Fig. 3, explain unlabeled bands such as in Fig. S7c).

These have been added.

11. I would personally suggest increased accessibility of data both by providing referenced anecdotal experiments (i.e. triplicate gels) in the supplemental information and by publishing the raw mass spectrometry data (in a repository) for access by readers.

The replicate results have been included in Fig. S11-S14. MS data will be uploaded to PRIDE database.

12. A higher resolution/reformatted version of figure S1 would be beneficial as the text is quite small and difficult to read in its current state.

We have reformatted Fig. S1. The color of the text has been changed to black for better representation.

Overall, the work presented has potential to integrate new and diverse methods, AlphaFold prediction and HDX-MS, to both the structural biology and wider scientific audiences. The concept of truncating the AMBRA1 protein by removing supposed IDRs to solve parts of its structure by cryoEM will also significantly contribute to the understanding the ubiquitin conjugation system and its complexes. However, in its current state, I believe there are several additional data analyses that are needed to sufficiently support the claims made in the manuscript.

We very much appreciate the referee's suggestions.

Reference:

1. Walters, B.T. et al. (2012) "Minimizing back exchange in the hydrogen exchange-mass spectrometry experiment," Journal of the American Society for Mass Spectrometry, 23(12), pp. 2132-2139. <https://doi.org/10.1007/s13361-012-0476-x>.

Reviewer #2 (Remarks to the Author):

The primary purpose of your review is to provide feedback on the soundness of the research reported. This will help authors to improve their manuscript and editors to reach a decision. When composing your report, the following questions might assist you in writing a well-justified review, but please feel free to raise any further questions and concerns about the paper.

- What are the noteworthy results?
- Will the work be of significance to the field and related fields? How does it compare to the established literature? If the work is not original, please provide relevant references.
- Does the work support the conclusions and claims, or is additional evidence needed?
- Are there any flaws in the data analysis, interpretation and conclusions? Do these prohibit publication or require revision?
- Is the methodology sound? Does the work meet the expected standards in your field?
- Is there enough detail provided in the methods for the work to be reproduced?

Where applicable, reporting summaries are requested from the authors to improve the transparency and reproducibility of published results. We hope the file, if included, will aid in your evaluation of the paper as they contain key information pertaining to study design and analysis.

To increase the transparency and openness of the reviewing process, we support our reviewers signing their reports to authors if they feel comfortable doing so. If, however, you prefer to send an anonymised report we will continue to respect and maintain your anonymity. The comments to the authors are subsequently shared with the other reviewers, but your identity will not be revealed unless you have signed your report.

Key results:

The authors isolate and structurally characterize an engineered mutant of AMBRA1, a tumor suppressor, in complex with DDB1, an adaptor that is part of the ubiquitination pathway. The portion of AMBRA1 used in this study includes an N-terminal helix-loop-helix motif which the authors show to be essential for binding to DDB1, and a non-canonical WD40 domain comprised of both an N terminal and a C terminal segment of AMBRA1, with the long intrinsically disordered region between those segments deleted. The authors fit an existing DDB1 model and an AlphaFold model prediction of the AMBRA1 WD40 domain into their 3 Å EM density map and refined the coordinates. Their EM/AlphaFold modelling observations are consistent with their HDMX observations. The authors conclude that, when the residues observed to be in the binding interface of their protein model were mutated, the mutations resulted in loss of function in in vitro (pull-down, ubiquitination) and in vivo (immunofluorescence) experiments.

Overall Feedback:

Because of the role of AMBRA1 in multiple pathways including cancer, this work is likely to be important for the field. However, the introduction and parts of the discussion are difficult to follow. There are many components to this system. In the introduction, the authors introduce about 15 different components. For scientists outside this field, it is unclear how these components interact and what roles they play. Perhaps it would be helpful to include a schematic or cartoon representation of each of the parts and how they interact / what role they play. The introduction may need to be expanded to more clearly define the components and what is important for the reader to know about them in order to understand the main findings of this study. At minimum, the introduction could be broken up into distinct paragraphs so that the reader has a clear indication of when a different role of AMBRA1 is being described. On a related note, there are times the authors refer to a substrate and it is not clear to me what substrate the reader is meant to understand.

We have now modified the introduction section and included a schematic representation for the key protein complexes in this study. AMBRA1 is able to bind and recruit large number of

proteins (substrates) for ubiquitination by CULLIN4 E3 ligase. Therefore, in introduction part we often generalise. However, in context of experiments substrate always referees to Cyclin D1. The autophagy part has now been removed per the reviewer's suggestion.

Additional data are needed to fully characterize the EM results presented. The authors state that their reconstruction is "around 3 Å" resolution, but also describe that one portion of the map (BPB domain, line 180–182) is quite flexible. It would help the reader if the authors include a map of local resolution to more clearly display the quality of the reconstruction which, as with most EM reconstructions, varies across the map. Additionally, the map appears to display some anisotropy, suggesting a preferred orientation of particles. It would be helpful for the authors to also include an orientation distribution plot so the reader can understand whether or to what degree a preferred orientation was observed.

The local resolution for the reconstruction and orientation distribution plot have now been included in Fig. S5f and Fig. S5g. Word "around" was removed and resolution is reported based on gold standard FSC.

There are some grammar choices that negatively impact the clarity of the writing. I am noting some of those which are most confusing. There are other instances of errors in subject-verb agreement and use of plural or singular nouns throughout the manuscript which are not captured here.

Thanks for your suggestions. We have sent our revised manuscript to language editing service before submitting for revision.

Specific points

- Line 37–38, What is meant by "highly" intrinsic disorder?

AMBRA1 is not completely disordered, but contains long disordered regions. Therefore "highly" was used. We have rephrased this sentence.

- Line 46 – 48 – the italicized portion is unclear "AMBRA1 uses its N-terminal helix-loop-helix and WD40 domain to bind the double-propeller fold of DDB1, whereas different regions target the specific cellular substrates for ubiquitination." Different regions of what? What are the specific cellular substrates?

We have revised this section.

- Line 51 – subject-verb agreement error – "these results...suggest"

This has been modified.

- Line 74 – There is not a clear connection between the first and second sentence.

We have now included more description.

- Lines 87 – Which protein is "its" referring to? "In nutrient-rich status, mTORC1 inhibits AMBRA1 by phosphorylation on its Ser-52 [6, 7]."

"Its" refer to AMBRA1 Ser52. We have revised this sentence.

- Line 89 – What is meant by "In addition to ULK1..."? This sentence structure is confusing. If "AMBRA1 functions as a substrate receptor for Beclin1 ubiquitination," what is the substrate in this reaction? A substrate of what enzyme? This is an example where I think a schematic figure may help.

Thanks for the suggestion. As AMBRA1 is involved in multiple complex biological signalling, therefore we agree that a figure will help the audience for better understanding.

- Lines 90 – 93 mention Beclin1 multiple times without defining what it is. Please also define PI3KC3 (line 92), CDK4/6 (line 96), and D-type cyclins (lines 94, 95). This section is an example of where it is unclear to the reader how these different components of the system interact.

Definition for Beclin1, PI3KC3, CDK4/6 and D-type cyclins are now included in the manuscript.

- Line 106 – “suggests a mechanism by which the AMBRA1 targets proteins involved in essential biological processes” Might the authors narrow this statement? This work goes on to compare structures of other other target proteins, but all within the same family, with very similar structures.

We have rewritten the sentence.

- Line 114 – “AMBRA1 has been characterized by WD40 domain in the N-terminal region while the rest of the protein is predicted as intrinsically disordered regions (IDR) (Fig. 1a) [12].” This wording is unusual and unclear. If a protein has been characterized by X, I would expect X to be a technique such as cryoEM or crystallography or even by functional studies. Perhaps AMBRA1 is predicted to have a WD40 domain, in which case I would still ask what this prediction is based on. In my effort to understand what was meant by this statement, I see that this language is very close to that used in reference 12, which is a review that cites for this statement, another review. Has a WD40 domain been observed in this protein? A homologue? Many homologues?

We have edited the sentence. It was reported that AMBRA1 has WD-40 repeats-region at the N-terminus (residues 1-175) by sequence analysis. The protein is not present in lower eukaryotes and highly conserved in vertebrates (PMID17589504, PMID36237336).

- Fig.1 -
 - o Panel b, what is the middle lane? Please label the lanes.

This middle lane is the purified MBP-tagged AMBRA1 protein, and now is labelled in the figure.

- o Panel c figure legend – what is meant by the “calculated central residue of an individual peptide?”

The central residue of a peptide refers to median residue number of peptide.

- o Panel d – What is the bottom row? I think it makes sense to swap the top and middle rows so the reader sees the AMBRA1 alone data first, and then AMBRA1 + E3 Ligase. This would be more consistent with how the figure legend and the main body text are written.

We are sorry for this mistake, the bottom row is the undeuterated apo AMBRA1. We have now swapped the top and middle panels, as well labelled the bottom row in the revised Fig 1.

- Fig. S1 – I cannot read any of the grey text in Fig. S1. It is too small.

This has been modified.

- Fig. S2 – What is the top row of mass spectra? Is this time zero? Please label. Would it make sense to use panels here for the bimodal isotopic envelopes (panel a) vs the kinetics

(panel b), vs the structure (panel c)? In the structure, please define the colors used (red, blue, grey).

The top row represents the undeterurated AMBRA1. We have added the label in the figure, arranged the panels and defined the three colors in the structure.

- Results lines 110–160. Do the open/closed states observed by HDX–MS correspond to folded/unfolded structures?

The open conformation results from a cooperative unfolding event simultaneously exposing multiple residues to solvent – unfolded structure. The closed one is folded.

- Figure 2 –

- o Panel C – The coloring is different in the model shown in the EM map as opposed to the model depicted on it's own. The two shades of green and the two shades of blue are very difficult to distinguish.

We have readjusted the figure and increased transparency of the map (map colour is gray). We have also changed the colors for the individual domains for better presentation.

- o Panel D –

- ♣ Please describe why these sections are highlighted. Does the left panel show the hydrophobic interactions and the right show the ionic interactions?

These two panels are showing the main interaction between AMRBA1^{WD40} and DDB1, and the key residues mediating the interaction are further mutated and tested for the binding, the ubiquitination and cell experiment.

- ♣ The text (lines 208–209) identifies highlighted residues from BPC including A381. The figure legend identifies BPC as being colored in blue. The left panel shows A381 but it is colored green. The text and figure appear inconsistent with each other.

We have corrected it.

- Figure 3 –

- o Panel b – In general, these bands are quite faint. Could the contrast be increased? Alternatively perhaps run the gels with more sample loaded. Why was the delta19 variant included?

We have adjusted contrast. Delta 19 variant was included to test importance of N-terminal helix for DDB1 interaction.

Panel c – why are there bands observed in the pull down that are not observed in the input?

The proteins are presented in the input. We have repeated the experiment and replaced with a more clear SDS-PAGE.

- o Panels d, e, and f – Please label all the observed bands. It is not clear, in panel e in particular, where there are multiple bands present in conditions the authors explain have no ubiquitination activity.

Thank you for this suggestion. These bands correspond to residual undigested GST-Cyclin D1 fusion protein (used for affinity purification). The figure was labelled accordingly.

- Line 175 – As noted above, more detail is needed with respect to global and local resolution as well as preferred orientation.

We have included the local resolution and particle orientation plot in Fig. S5.

- Fig. S3.
 - o Panel c – please include how many particles were included at each step and resolution at each step.

This has been included.

- o The GSFSC and the portion of the EM map shown at the lower right need panel labels and descriptions in the figure legend.

These have been included.

- Fig. S4.
 - o Panel a – I'm used to seeing a map vs model as one curve, not two separate curves. If we are to compare these two curves, they look pretty different.

Such curves may be contributed by the dynamic of DDB1 and AMBRA1. Consistently, previous reports (PMID: 36378961, 35101445 etc.) and we showed that DPB domain of DDB1 is highly flexible relative to the other parts of the protein, and our final reconstruction is a combination of multiple orientations of this domain, therefore our model of this part could not meet the satisfactory correlation to the map. We have tried to mask out the BPB domain for refinement, however, the quality of the map became worse as the size of the protein gets smaller, therefore we did not continue to persuade .

- o Panel c – The smaller portions appear to be chosen to show side chain densities, and the larger portions to show whole domains. The larger domain views are too busy to make out details. Consider showing where in the domain structures the smaller highlighted portions are taken from.

This has been edited.

- Fig. S5. Please describe the figure in the legend. Are these different classes in 3D classification? Something else? How many particles are present in each class? 3D classification is mentioned in the main text, but it is not shown as part of the EM processing workflow (Fig. S3)

We have now included more details about this processing task in the figure legend. We input the 780,099 particles from NU-refinement in cryosparc and imported it into Relion 3 for 3D classification without alignment. From this job, 5 classes were generated, which can clearly showed that the BPB domain adopts different orientation. We did not include this in the EM processing workflow as we mainly used the reconstruction before this analysis. The reason is that we have tried to refine the individual maps but this led to worsen map quality, therefore we decided to report the one from previous job.

- Lines 206 – 209: The N-terminal helical motif is almost entirely engulfed by the large pocket formed between BPA and BPC double-propeller of DDB1, stabilizing between the hydrophobic interactions contributed by W953, F1003, V360, L814, A381 of BPC domain with V10, L13, W14 in the first helix.

- o “stabilizing between” is confusing. Perhaps the authors mean stabilized by?

o At the end of the sentence, “in the first helix” seems incomplete. Please state which protein is meant by this clause, possibly “in the first helix of AMBRA1.”

These have been edited.

- Lines 213 – 215: We incubated purified MBP tagged AMBRA1FL, AMBRA1NΔ19 or AMBRA1V10R/L13R/W14R and AMBRA1V10R/L13R/W14R/H929A with His6 tagged DDB1.

o The “and” is confusing. Seems to imply that AMBRA1V10R/L13R/W14R/H929A was included in all experiments with one of the other variants.

We have changed ‘or’ to ‘;’.

- Lines 220 – 223 has grammar errors that make it difficult to understand.

o As written: Comparison of our AMBRA1–DDB1 structure to other DDB1 complex, the binding between DDB1 and AMBRA1WD40 is similar to other adaptors bound to DDB1 including DCAF1 (PDB:6ZUE), Simian virus 5V (PDB: 2B5L), CSA (PDB: 6FCV) · DDB2 (PDB: 3EI4), suggesting a common molecular mechanism of DDB1 recognition (Fig. S6b).

o Potential revision: Comparison of our AMBRA1–DDB1 structure to other DDB1 complexes shows that the binding between DDB1 and AMBRA1WD40 is similar to other adaptors bound to DDB1 including DCAF1 (PDB:6ZUE), Simian virus 5V (PDB: 2B5L), CSA (PDB: 6FCV) · DDB2 (PDB: 3EI4), suggesting a common molecular mechanism of DDB1 recognition (Fig. S6b).

We very much appreciate the referee's suggestions. Text was revised.

- Fig. S6

o Panel B – the referenced structures are all of an adaptor–DDB1 complex. The figure overlays the structures of the different adaptors. It appears there is only one model displayed for the DDB1. Is the DDB1 structure exactly the same in all of these complexes?

Yes, we included only one DDB1 in order not to complicate the figure.

- Lines 226 – 232 has many grammar errors that are hard to understand. The below has possible revisions. This section also asks the reader to visualize movements within the proposed structural model but, again, there are many components involved. Could the authors present this in a schematic or even a movie?

o Our model revealed a ~70 Å distance between RBX1 and central cavity of AMBRA1. Notably, the 3D classification without alignment analysis confirmed the plasticity of BPB domain of DDB1, which can adopt diverse orientations to position AMBRA1 for substrate (what is the substrate?) ubiquitination. RBX1 is likely to reposition as a result of neddylation, therefore establishing an open active conformation to mediate transfer of ubiquitin from E2 to substrate, which shortens the distance between the substrates and the enzymes.

We have edited the sentences in the manuscript.

- Line 247: AMBRA1 functions as a substrate receptor to recruit Cyclin D1.

o What is Cyclin D1 recruited to? What is the substrate?

Cyclin D1 is the substrate of AMBRA1.

- Line 247 – 249: Therefore, disrupting the interaction between AMBRA1 and DDB1 would fail to mediate the ubiquitination of its substrate.

o This sentence is confusing. What is doing the mediation? What does “its” refer to?

The interaction between AMBRA1 and DDB1 is important since AMBRA1 bridges DDB1 and Cyclin D1, therefore Cyclin D1 (substrate) can be ubiquitinated.

- Lines 249 – 256 The authors mention poly-ubiquitination twice, note conditions in which they “did not observe any ubiquitination” and also mention “ubiquitination.” Is there an important distinction to be made between ubiquitination and polyubiquitination?

We are sorry for our inconsistency within the manuscript. We have corrected the words.

- Lines 252 – 256 We further tested the effect of AMBRA1 mutants including AMBRA1V10R/L13R/W14R and AMBRA1V10R/L13R/W14R/H929A in promoting the ubiquitination of Cyclin D1-CDK4 (Fig.3f), none of these mutants execute the activity, indicating the N terminal helix-loop-helix of AMBRA1 is essential for association with CRL4, which in turn mediate the polyubiquitination of Cyclin D1.

o The sentence makes a conclusion about the N terminal helix-loop-helix domain, but the H929A mutation is not in that domain.

Yes, our data showed that the critical interaction is mediated by the N terminal helix-loop-helix of AMBRA1, therefore V10R/L13R/W14R was sufficient to disrupt the binding.

- Line 263 – What is LC3-II and why is it important? How is LC3-II involved in autophagy?

LC3-II is a standard protein indicator for autophagosomes. It is generated by conjugating the cytosolic LC3-I to phosphatidylethanolamine (PE) on the nascent autophagosomes. Conversion of LC3 (LC3-I to LC3-II) is a clear indication of autophagy and commonly used within the field. Based on other reviewer suggestion, we have performed more relevant cell cycle progression experiments and autophagy part is now removed.

- Line 276–277 This was further supported by LC3 immunoblotting results showing that AMBRA1 can induce autophagy by interacting with CRL4 E3 ligase. (Fig.4d–e).

o E3 ligase is not included in the figure mentioned in this sentence. What is the connection?

CRL4 E3 ligase is CULLIN4-RBX1-DDB1-AMBRA1 complex. Autophagy part is now removed.

- Fig. 5 – This figure needs a more complete description in the figure legend. There are grey portions shown in surface and in ribbon model, and multiple different colors used in the ribbon model. Is any experimental evidence from this paper used in this figure, or are all the structures shown AlphaFold2 predictions and the PDB 2HYE structure used in Figure S6?

This has been added.

- Fig. S7

o Panel b – what is the band ~70 kDa?

The ~70 kDa band is heat shock protein 70, a protein commonly copurified from protein purification.

o Panel d – There are 2 bands. I can't tell if the label is one label or two labels. The bottom line of the label is missing the “-“ which is present in the other labels on this figure indicating the position of the band corresponding to the label.

The label has been added.

- Methods

o Line 662. This portion begins with incubating the “cell supernatant” with amylose beads. There is no mention to the lysis and centrifugation steps that, in the previous paragraph, precede this step. Please include all steps of this purification.

This has been added.

o Line 666 – What is meant by “Full length AMBRA1 NΔ19?” How is the construct both full length and a truncation?

This has been changed to AMBRA1 NΔ19.

Reviewer #3 (Remarks to the Author):

Liu et al characterized the structure of the AMBRA1 beta-propeller, focusing on its interaction with DDB1, which is functional to Cyclin D1 ubiquitylation. The relevance of this pathway is highlighted by three papers recently published on *Nature*, showing the importance of the AMBRA1/DDB1/CyclinD1 axis in cancer and neurodevelopment. However, although of potential interest, a series of findings in the article make it not entirely novel or sound. Moreover, the intriguing and potentially groundbreaking findings regarding AMBRA1 structure are not fully balanced by any complete associated functional characterizations of the underlying biological mechanisms.

Despite the claims of the authors in the abstract (line 42–43: “Here we present the cryo-EM structure of AMBRA1 in complex with DDB1 at 3 Å resolution”), the structure itself is far from being totally understood, since they describe only a portion of the protein whose function has been already characterized by others (PMID 25499913; PMID 33854232), who identified the region for binding with DDB1 in the first 180 residues and the relevance of this region for Cyclin D degradation.

Thank you for your comments. These are important milestone papers whose observation we have now been able to show at the structural level.

Also, the statement regarding the hypothesized WD40-N – WD40-C folding (lines 135–141) is misleading and should be removed or rephrased. Last, put in this manner, the reader may think that this is a totally novel concept, but the exact same conformation was already proposed and published this year (PMID 36243772, not cited in the manuscript), taking advantage of the AlphaFold2 prediction. Of note, the results here cited are coherent with the experimental data provided by the authors.

PMID 36243772 was published during our manuscript preparation and we unfortunately missed it. The reference is now included in the manuscript.

The title as well is not fully appropriate. Despite in Fig. 3d the authors evaluate the ability of wildtype and mutant AMBRA1 to recruit and promote cyclin D ubiquitination, there is no structural data regarding AMBRA1 substrates (e.g., cyclin D) recruitment, since the manuscript is largely focused on the upstream AMBRA1–DDB1 interaction.

We understand and appreciate the reviewer's concern here. Our reason is that AMBRA1-DDB1 complex represents structural platform for recruitment of many substrates including the recently identified Cyclin D1. In addition, we provide biochemical characterization of Cyclin D1 binding and show that WD40 domain alone is sufficient to mediate DDB1-AMBRA1-Cyclin D1 interaction and in vitro ubiquitination assay. Of course, the structure of this complex would provide highly valuable information, as would structures of AMBRA1 with other substrates. Since AMBRA1 interacts with many substrates, one may expect that interaction interfaces will differ as well. But in all these cases AMBRA1 WD40/DDB1 structure will serve as base for substrate requirement to the E3 active site. If the reviewer is still of opinion that title is not appropriate – we would be prepared to alter it.

On a positive note, I found extremely intriguing the multiple AMBRA1WD40 single and multiple residues mutants. Are they mutated in cancer? Do they have any effect on cell cycle regulation? What about their sensitivity to CHK1 inhibitors? Could the authors better define what are the physio-pathological consequences of this mutants apart from cyclin D1 ubiquitination?

Thank you. Based on <https://cancer.sanger.ac.uk/> database search we did not find these mutations in cancer. As suggested, we performed cell cycle progression experiments and could see similar phenotype as described previously. Namely, in cells expressing AMBRA1 point mutations we have observed significantly increased percentage of cells in S phase. These mutations would probably promote tumor formation and the invasiveness of cancer through Cyclin D1/CDK pathway. From another perspective, the AMBRA1 WD40 mutants target AMBRA1-DDB1 interaction and this could impact all CULLIN4-AMBRA1 related functions including autophagy. One may expect range of defects not limited to some specific substrate or pathway as may be the case with some of the identified cancer mutations. However, at this stage we can only speculate about pathological consequences.

Being the AMBRA1WD40 able to interact per se with its substrate cyclin D1 (as demonstrated in Fig. 3c), one may suppose that also other substrates of its scaffolding activity for CRL4 ubiquitylation could interact similarly. Consequently, it would be interesting to define its interactome to identify novel targets of its DCAF activity, while potentially excluding the plethora of other AMBRA1 interactors.

This excellent question and proposal, and we plan to address in a separate study utilizing proximity labeling tools.

Additionally, the manuscript contains a series of flaws here listed:

– Liu et al provide a further structural characterization by narrowing down the interaction region to the first 49 residues plus some others present in the C-Term WD40 domains. This has been demonstrated through cryoEM experiments, in vitro binding, and ubiquitylation assays carried with an NΔ49 mutant. However, the NΔ49 deletion would likely disrupt the local propeller structure, and it is not a proof of direct binding.

We have done control binding experiment with WD40N-construct without the C-terminal propeller part. This construct was able to form complex with DDB1 upon co-expression despite lacking half of the WD40 domain. This together with NΔ49 mutant demonstrates centrality of AMBRA1 N-terminus for DDB1 interaction. Perhaps more importantly, we have now done

binding experiments with series of point mutants which are less likely to disrupt the propeller structure.

- The authors state that the residues H929 and D866 in the C-TERM WD40 could mediate ionic bonding with DDB1, but this is not experimentally validated with a sufficient level of detail. In Fig 3b the authors investigate the AMBRA1 V10R/L13R/W14R/H929A and AMBRA1V10R/L13R/W14R mutants binding with DDB1. However, the AMBRA1V10R/L13R/W14R mutations already completely abolish the interaction, making it difficult to determine the influence of the H929A substitution on the interaction. A single mutant H929A should be done to evaluate the impact of the residue for AMBRA1/DDB1 binding.

Moreover, these multiple mutations on AMBRA1 could disrupt the local propeller structure rather than impacting the DDB1 interaction.

As suggested, we have now preformed experiments using single mutants.

This aspect should be carefully checked, also considering that in Fig. 3b the mutant proteins seem to be less expressed as a possible result of destabilizing mutations.

Fig. 3b was not sufficiently explained and we apologize for this mistake. In this experiment proteins are not co-expressed, they were expressed separately and mixed together prior to pull down assay. We have repeated the experiment and made sure all inputs are at the same level.

- The experiments suggest the importance of the N-TERM helix loop helix in AMBRA1 DDB1 interaction but do not entirely rule out or fully characterize the importance of other regions in the DDB1-AMBRA1 binding.

We now include single mutants from both parts of WD40 domain and characterize their contribution to AMBRA1-DDB1 interaction and in vitro ubiquitination.

- In Fig. 4 the authors investigate the role of AMBRA1 depletion in autophagy through reconstitution experiments with a wild type or NΔ49 mutant. This part of the manuscript is definitely underdeveloped with respect to the previous sections and does not add anything novel enough. Indeed, the methods used in this study are not rigorous nor up-to-date, therefore it is impossible to draw any conclusion from the results obtained. Figures 4a-c need a proper autophagic flux assay, that typically include starvation/cloroquine or bafilomycin (as the authors do in Fig. 4d). In Fig. 4d the endogenous AMBRA1 signal is not visible so it is impossible to assess the efficiency of the AMBRA1 interference. Again in Fig. 4d, to prove the efficacy of Bafilomycin treatment unprocessed LC3 I signal is necessary. Moreover, since the manuscript focuses on the structural characterization of AMBRA1 beta-propeller, DDB1 interaction, and Cyclin D1 ubiquitylation, the most correct in-cell validation should focus on the control of cell cycle progression and its consequence on genome instability.

These assays were focused to basal levels of autophagy in HEK293 cells, and bafilomycin was used for all experiments. Granted, starvation experiments would largely strengthen our claims. However, we agree with reviewer comment that the most correct in-cell validation should focus on the control of cell cycle progression. Therefore, we have decided to temper our claims about autophagy and instead performed cell cycle progression experiments using AMBRA1 point mutants identified in this study.

REVIEWER COMMENTS

Reviewer #1 (Remarks to the Author):

Dear authors,

In this manuscript you have used three techniques to characterize AMBRA1 with relevance to its role in substrate recruitment effecting autophagy and UPS. (1) The HDX MS study shows the dynamics of the entire AMBRA1 protein. The flexibility in the extended loop of Blade IV is well shown and the EX1 kinetics favoring the closed state when E3 is present support the stabilizing effects of the E3 ligase. (2) The cryo-EM of AMBRA1-WD40 supports the split domain concept and shows that the AMBRA1 protein still folds to its distinct B-propellor shape without the Blade IV extended loop. (3) A series of pull down and activity experiments highlight critical residues for complexing with DDB1 and subsequent ubiquitination activity and cell cycle progression.

Overall, the manuscript has been greatly improved. I appreciate the time taken to address and respond to both my and the other reviewers' comments. The manuscript reads much more clearly, and I very much appreciate the more in-depth hdx analysis, single mutant study, cell cycle study, and addition of figures, especially Figure 1a, S2, S4, etc. I have a few minor points just to polish the manuscript.

Minor points

1. In the abstract Line 40, saying that HDX analyzed "the domain organization" is a bit misleading, I might just mention the dynamics.
2. Figure 1B is still missing lane labels. All of the other gel images have been labeled appropriately.
3. Line 149 mentions "Secondary structure and domain prediction" but there does not seem to be details on how this prediction was carried out or how it could be reproduced. Also, this sentence references Fig S1 (which is a great figure!) but doesn't make a lot of sense with this particular in-text sentence aside from maybe pointing out the black bars.
4. While it is mentioned in the methods that the AlphaFold model was downloaded, an accession number is still missing.
5. I think readability would be improved if the specific complex name was consistently used instead of referring to it more broadly as "E3 ligase complex" (i.e. Figure 1d).
6. Not critical, but the order that the subfigures in Fig 3 are laid out on the page doesn't really make sense (a, b, e ; c, d, f).
7. In Fig 4, I would bring the definition of siSCR to figure legend 4a (instead of 4b) since it is first referenced there.
8. I greatly appreciate the addition of Fig S2 per my suggestion. Just out of curiosity, are the IDRs moderate (yellow) stability what was expected? How do you explain the half helix that is highly deuterated (red) on the lower left of the figure?
9. Fig S8b is missing DCAF1 in the figure legend.

The authors have done important work on characterizing the dynamics of AMBRA1, a critical protein in cellular homeostasis and a hot topic in human health and disease. The abbreviated AMBRA1-WD40 construct capable of ubiquitination of cyclin D1 (though to a lower degree than the WT) allowed the authors to determine an $\sim 3\text{\AA}$ cry-EM structure of the highly structured regions of the protein and proves the presence of a split domain. I look forward to seeing how the authors continue to probe the mechanisms of this E3 ligase system - especially with the proximity labeling mentioned in the response to reviewers!

Reviewer #2 (Remarks to the Author):

I find the revised manuscript greatly improved. The addition of the schematic diagram in Figure 1a is particularly helpful. The clarity of the writing is significantly improved. The inclusion of the cell cycle progression data is also a benefit.

One important point remains unclear. Is the "N terminal helix" or "N terminal helix loop helix" domain (both descriptions are used, sometimes in the same sentence as in Fig S3 figure legend) a part of the WD40 domain, or is it considered a separate domain? The authors may consider noting the location of this domain in the schematic now included in Fig 1a. Lines 224-225 note that the helix-loop-helix motif is separate from the "globular core WD40 domain."

There are additional minor points arising from the revisions. Many are related to figures. In general, main text references to figures and the details of the figure should be rechecked.

- Line 151 references Fig S1. Is this correct? Fig S1 shows HDX-MS results but this sentence is about secondary structure and domain predictions.
- Line 191 contains the first mention of Fig 3 and none of the 6 panels are specified. All 6 panels do not relate to this statement. (If they do, please explain in the text how or why.) It would be more helpful for the authors to direct the reader to the relevant panels.
- Line 198 refers to Figure S6. In my understanding, this statement is relevant only to panel c in this figure. Panels A and B are not mentioned or discussed in the text. Panel B indicates a 3.2 Å resolution but the main text, in the sentence referencing this figure, claim a 3.08 Å resolution.
- Line 210 discusses the N terminal helix loop helix domain made up of residues E6-K41. Again, I think including this in the schematic in Fig 1a or the structure shown in Fig 2c would be helpful. It also seems to me that perhaps Fig S3C indicates this region (shown in red) and could be referenced here for improved clarity.
- Line 236 – Authors have not explained why they use a delta19 mutant here instead of the delta39 mutant explained and used previously.
- Between lines 284 and 293, supplemental figures 10, 11, 12, and 13 are not discussed. These supplemental figures are referenced in the methods but are out of order. The numbering is confusing and distracting. Line 884 references Fig S11 and S13 and then line 952 references Fig S12 Please reorder in the figures in the order discussed in the text.
- Line 347 – Do the authors mean "substrate recruitment" (rather than "substrate requirement")?
- Figure 1d – the numbers on the x axes are too small and difficult to read.
- The layout of the panels in Figure 3 is confusing. Left to right the panels are a, c, b, d, e, f. Top to bottom they are a, b, e, c, d, f. They are also not presented in the figure in the order they are discussed in the text. Is the band at 70 kDa in panel d, e, and f the Hsp70

contaminant again? As currently labeled, it looks like it Ub-cyclin D1, but it appears to be present in negative controls.

- In Figure 4d, the colors are indistinguishable and should be revised for more contrast. SiSCR is used in 4a but only defined in the 4b (and not 4a) figure legend.
- Line 777 – In the materials and methods for co-expression purification, the authors state that “wash buffer” was used without indicating details. This is unclear because the paragraph before and the paragraph after each describe a “wash buffer” for other purifications and those buffers, though similar, are not the same (lines 763 and 789).
- Line 796 - Bold “Fig S10” for consistency.

Supplemental tables and figures

- Table S2
 - o Electron exposure ($e^-/\text{\AA}^2$) is given as 1.1792 which seems very low. The methods section reports 58.96 $e^-/\text{\AA}^2$ which seems reasonable.
 - o The map resolution range is listed as 30 – 3.08 Å but are not shown any maps with ~ 30 Å resolution.
- Fig S2 – Please also show confidence of AlphaFold prediction. How confident are we of the loops? This is important with respect to the mapping of various binding sites in Fig 4e.
- Fig 34 – All of the text and numbers on the axes and legends of the panels are too small to read. Could this be split to cover 2 pages?
- Fig S5- In panel e – please state why this particular portion of the map is shown. In panel f – thank you for including the local resolution data. Additional views need to be shown as well (at least 2 total views, preferably 3) in order to show all parts of the map. The text on the color legend is also much too small to read.
- Fig 6 – Panels a and b do not appear to be discussed in the text.
- Fig S12 needs labels on the individual panels for the relevant bands, as was included on the other similar figures.
- Fig S14 needs a lot more labeling. Molecular weight markers are not labeled. The relevant bands are not labeled. Some gels or membranes are labeled 1 – 4, others 1-6, a couple are not labeled at all. In the top right set of membranes, there is text written across some of the data.

Reviewer #3 (Remarks to the Author):

After this author revision, my concerns about manuscript novelty do, indeed, remain. The paper merely builds upon previously published work. Specifically, the study primarily validates the structure of the beta-propeller domain of AMBRA1, as demonstrated in PMID 36243772, and characterizes the DDB1-AMBRA1 interaction, the region of which has already been established and studied. The paper here cited shows, indeed, the beta-propeller domain of AMBRA1 and is mentioned in reference 17, although not thoroughly discussed, leaving unclear to which extent the present results are in line with previous evidence. Rather, the recommended interactome analysis of the AMBRA1 WD40 would have introduced novelty to the manuscript; however, unfortunately, the authors decided to not address this point.

Furthermore, I still find the title "Structural basis for substrate recruitment by AMBRA1 E3 ligase receptor" somehow misleading, as it implies the characterization of a universal binding mechanism. At variance with that, the authors focus only on the structural characterization of the DDB1-AMBRA1 binding (not novel).

Moreover, the required and performed additional experiments yield inconsistent results. Although the authors carried out experiments with single mutants, as suggested, the mutants V10R and L13R disrupt the binding with DDB1, while mutants W14R and H929A still exhibit interaction (Fig S9a). It remains unclear why single mutants show an impact on Ubiquitination experiments (S9b) and on functional assays (Fig 4d), and this discrepancy is not addressed at all in the manuscript.

Additionally, despite my suggestion to conduct more accurate experiments on autophagy, the authors have instead completely removed this section and focused only on a minimal set of experiments related to cell cycle. As for those, the authors included a preliminary analysis of the cell cycle phase distribution, which also misses a proper control (i.e., cells not interfered for AMBRA1 – Fig. 4d). The authors have also ignored the request to investigate the effects on genomic stability and the potential effect on CHK1 inhibition sensitivity, which is a significant oversight.

Last, the majority of the new supplementary figures are inadequately prepared, lacking proper labeling for different panels, and exhibiting unexplanatory figure legends.

REVIEWER COMMENTS

Reviewer #1 (Remarks to the Author):

Dear authors,

In this manuscript you have used three techniques to characterize AMBRA1 with relevance to its role in substrate recruitment effecting autophagy and UPS. (1) The HDX MS study shows the dynamics of the entire AMBRA1 protein. The flexibility in the extended loop of Blade IV is well shown and the EX1 kinetics favoring the closed state when E3 is present support the stabilizing effects of the E3 ligase. (2) The cryo-EM of AMBRA1-WD40 supports the split domain concept and shows that the AMBRA1 protein still folds to its distinct B-propellor shape without the Blade IV extended loop. (3) A series of pull down and activity experiments highlight critical residues for complexing with DDB1 and subsequent ubiquitination activity and cell cycle progression.

Overall, the manuscript has been greatly improved. I appreciate the time taken to address and respond to both my and the other reviewers' comments. The manuscript reads much more clearly, and I very much appreciate the more in-depth hdx analysis, single mutant study, cell cycle study, and addition of figures, especially Figure 1a, S2, S4, etc. I have a few minor points just to polish the manuscript.

Minor points

1. In the abstract Line 40, saying that HDX analyzed "the domain organization" is a bit misleading, I might just mention the dynamics.

We have removed the "the domain organization" in Line 40.

2. Figure 1B is still missing lane labels. All of the other gel images have been labeled appropriately.

We have now labelled all the bands in the gel (Fig.1b).

3. Line 149 mentions "Secondary structure and domain prediction" but there does not seem to be details on how this prediction was carried out or how it could be reproduced. Also, this sentence references Fig S1 (which is a great figure!) but doesn't make a lot of sense with this particular in-text sentence aside from maybe pointing out the black bars.

We have now included the secondary structure and domain prediction for AMBRA1 protein which were performed using PSIPRED and WDSPPdb2.0 (now Fig. S2).

4. While it is mentioned in the methods that the AlphaFold model was downloaded, an accession number is still missing.

The identifier AF-Q9C0C7-F1 for the AlphaFold model has been added in the Materials and Methods section.

5. I think readability would be improved if the specific complex name was consistently used instead of referring to it more broadly as "E3 ligase complex" (i.e. Figure 1d).

We have change E3 ligase complex to Cullin4A-DDB1-RBX1 in Fig. 1d, Fig. S4 and Fig. S5.

6. Not critical, but the order that the subfigures in Fig 3 are laid out on the page doesn't really make sense (a, b, e ; c, d, f).

We tried to adjust the figure in the order (a, b, c, d, e, f), however, in such arrangement, the label of b panel is too small to read, but now we have done our best to readjust the figures in a more reasonable order.

7. In Fig 4, I would bring the definition of siSCR to figure legend 4a (instead of 4b) since it is first referenced there.

We agreed. We have added the definition of siSCR (scrambled siRNA) in figure legend 4a.

8. I greatly appreciate the addition of Fig S2 per my suggestion. Just out of curiosity, are the IDRs moderate (yellow) stability what was expected? How do you explain the half helix that is highly deuterated (red) on the lower left of the figure?

Thank you for pointing this out. We have extended the figure to contain both AlphaFold confidence measure- per-residue confidence score (pLDDT) and HDX heatmap for comparison. The WD40 core has high prediction score and relatively low deuteration uptake. In contrast, IDRs and helices contained within are predicted with low confidence. Therefore, one may argue that HDX in this case provides better picture and this helix is somewhat shorter than predicted by AlphaFold.

Perhaps this is good opportunity to stress that relative position of these helices in respect to WD40 domain is still an open question since they were not visualized experimentally.

9. Fig S8b is missing DCAF1 in the figure legend.

We are sorry and now has included DCAF1 in the figure legend (now Fig. S9b).

The authors have done important work on characterizing the dynamics of AMBRA1, a critical protein in cellular homeostasis and a hot topic in human health and disease. The abbreviated AMBRA1-WD40 construct capable of ubiquitination of cyclin D1 (though to a lower degree than the WT) allowed the authors to determine an $\sim 3\text{\AA}$ cry-EM structure of the highly structured regions of the protein and proves the presence of a split domain. I look forward to seeing how the authors continue to probe the mechanisms of this E3 ligase system - especially with the proximity labeling mentioned in the response to reviewers!

Thanks again!

Reviewer #2 (Remarks to the Author):

I find the revised manuscript greatly improved. The addition of the schematic diagram in Figure 1a is particularly helpful. The clarity of the writing is significantly improved. The inclusion of the cell cycle progression data is also a benefit.

Thank you for your advices and careful assessment of this work.

One important point remains unclear. Is the "N terminal helix" or "N terminal helix loop helix" domain (both descriptions are used, sometimes in the same sentence as in Fig S3 figure legend) a part of the WD40 domain, or is it considered a separate domain? The authors may consider noting the location of this domain in the schematic now included in Fig 1a. Lines 224-225 note that the helix-loop-helix motif is separate from the "globular core WD40 domain."

N terminal helix loop helix refers to E6-K41, which posits before the WD40 core. We have noted this part in the schematic (Fig. 1a) and corrected the figure legend in Fig. S3 (now Fig. S4) to N terminal helix-loop-helix as suggested.

There are additional minor points arising from the revisions. Many are related to figures. In general, main text references to figures and the details of the figure should be rechecked.

We have carefully gone through our text and figures, as well as made sure all the figures are correctly referenced.

- Line 151 references Fig S1. Is this correct? Fig S1 shows HDX-MS results but this sentence is about secondary structure and domain predictions.

We have now included the secondary structure and domain prediction for AMBRA1 protein which were performed using PSIPRED and WDSPPdb2.0 (now Fig. S2).

- Line 191 contains the first mention of Fig 3 and none of the 6 panels are specified. All 6 panels do not relate to this statement. (If they do, please explain in the text how or why.) It would be more helpful for the authors to direct the reader to the relevant panels.

We specify this statement to Fig. 3c-d because the purified protein AMBRA1^{WD40}-DDB1 was used for *in vitro* pull down and ubiquitination assay.

- Line 198 refers to Figure S6. In my understanding, this statement is relevant only to panel c in this figure. Panels A and B are not mentioned or discussed in the text. Panel B indicates a 3.2 Å resolution but the main text, in the sentence referencing this figure, claim a 3.08 Å resolution.

We have corrected this statement relevant to Fig. S7c and Fig. S6d. In addition, we have included the description about panel a and b in the Materials and Method “Atomic model building and refinement” section. Similar assessment was reported in PMID 32868926 and 31672913.

- Line 210 discusses the N terminal helix loop helix domain made up of residues E6-K41. Again, I think including this in the schematic in Fig 1a or the structure shown in Fig 2c would be helpful. It also seems to me that perhaps Fig S3C indicates this region (shown in red) and could be referenced here for improved clarity.

Per suggestion, we have included the notion for the helix-loop-helix in the schematic (Fig. 1a) and indicated it in the Fig.2c. We also referenced Fig. 1a and Fig. S3c (S4c now) in this sentence.

- Line 236 – Authors have not explained why they use a delta19 mutant here instead of the delta39 mutant explained and used previously.

Since the V10, L13, and W14 residues are located within the first helix, therefore we used delta19 mutant, ie removal of the first helix instead of deleting the entire helix-loop-helix.

- Between lines 284 and 293, supplemental figures 10, 11, 12, and 13 are not discussed. These supplemental figures are referenced in the methods but are out of order. The numbering is confusing and distracting. Line 884 references Fig S11 and S13 and then line 952 references Fig S12 Please reorder in the figures in the order discussed in the text.

We are sorry. We have arranged the last few supplementary figures in the correct order. These figures are the replicated experiment result for the assays in this work.

- Line 347 – Do the authors mean “substrate recruitment” (rather than “substrate requirement”)?

Yes, we have corrected the word.

- Figure 1d – the numbers on the x axes are too small and difficult to read.

We have increased the font size of the X axes.

- The layout of the panels in Figure 3 is confusing. Left to right the panels are a, c, b, d, e, f. Top to bottom they are a, b, e, c, d, f. They are also not presented in the figure in the order they are discussed in the text. Is the band at 70 kDa in panel d, e, and f the Hsp70 contaminant again? As currently labeled, it looks like it Ub-cyclin D1, but it appears to be present in negative controls.

We have re-adjusted the panels in Fig.3 in a more reasonable order. Regarding the band of 70 kDa in panel d-f, after carefully going through our data, we think it might be the uncleaved GST-myc-Cyclin D1, as it also showed up in the negative control lanes. Please check the Fig. 3c as it showed the migration position of GST-myc-Cyclin D1. We have included the description about this band in the figure legends.

- In Figure 4d, the colors are indistinguishable and should be revised for more contrast. SiSCR is used in 4a but only defined in the 4b (and not 4a) figure legend.

We have increased the contrast and size for better presentation in Fig. 4d (now Fig. 4b). In addition, we have introduced the definition of siSCR (scrambled siRNA) in figure legend for Fig.4a.

- Line 777 – In the materials and methods for co-expression purification, the authors state that “wash buffer” was used without indicating details. This is unclear because the paragraph before and the paragraph after each describe a “wash buffer” for other purifications and those buffers, though similar, are not the same (lines 763 and 789).

We have now included the composition of the wash buffer for each protein purification in the Materials and Methods section.

- Line 796 - Bold “Fig S10” for consistency.

Yes, the Fig S10 is bolded now (now Fig. S11).

Supplemental tables and figures

- Table S2

o Electron exposure ($e^-/\text{Å}^2$) is given as 1.1792 which seems very low. The methods section reports 58.96 $e^-/\text{Å}^2$ which seems reasonable.

We have corrected the dose.

- The map resolution range is listed as 30 – 3.08 Å but are not shown any maps with ~30 Å resolution.

We have removed this column.

- Fig S2 – Please also show confidence of AlphaFold prediction. How confident are we of the loops? This is important with respect to the mapping of various binding sites in Fig 4e.

We have extended the figure to contain both AlphaFold confidence measure-per-residue confidence score (pLDDT) and HDX heatmap for comparison.

- Fig 34 – All of the text and numbers on the axes and legends of the panels are too small to read. Could this be split to cover 2 pages?

We have now changed the labels of AMBRA1+E3 ligase to AMBRA1-Cullin4A-DDB1-RBX1 and splitted the figure into 2 pages.

- Fig S5- In panel e – please state why this particular portion of the map is shown. In panel f – thank you for including the local resolution data. Additional views need to be shown as well (at least 2 total views, preferably 3) in order to show all parts of the map. The text on the color legend is also much too small to read.

There is not specific reason showing this view for the panel e, therefore in order not to confuse the reader, we decided to remove it. Instead, we showed the local resolution data in 3 total views. The color legend is enlarged as well.

- Fig S6 – Panels a and b do not appear to be discussed in the text.

The description for these two panels is added in the Materials and Method section.

- Fig S12 needs labels on the individual panels for the relevant bands, as was included on the other similar figures.

The labels are included now.

- Fig S14 needs a lot more labeling. Molecular weight markers are not labeled. The relevant bands are not labeled. Some gels or membranes are labeled 1 – 4, others 1-6, a couple are not labeled at all. In the top right set of membranes, there is text written across some of the data.

All the panels in the figures are now labelled.

Reviewer #3 (Remarks to the Author):

After this author revision, my concerns about manuscript novelty do, indeed, remain. The paper merely builds upon previously published work. Specifically, the study primarily validates the structure of the beta-propeller domain of AMBRA1, as demonstrated in PMID 36243772, and characterizes the DDB1-AMBRA1 interaction, the region of which has already been established and studied. The paper here cited shows, indeed, the beta-propeller domain of AMBRA1 and is mentioned in reference 17, although not thoroughly discussed, leaving unclear to which extent the present results are in line with previous evidence. Rather, the recommended interactome analysis of the AMBRA1 WD40 would have introduced novelty to the manuscript; however, unfortunately, the authors decided to not address this point.

Thank you for your comments. Our work is providing experimental insights into structure and dynamics of the AMBRA1-DDB1 complex. This is the first structure of the complex and allows for mechanistic analysis at the residue level. This work will serve as foundation for further biochemical and structural studies focusing to diverse AMBRA1 substrates and may open a new avenue for development of small molecule drugs. Previous studies have resulted in important milestone papers whose observations we have now been able to show at the structural level. We thank the reviewer for helpful comments that considerably improved our manuscript.

We agree that comprehensive proteomics analysis would provide another dimension to this study. We were able to perform affinity-purification using AMBRA1 WD40 domain (the construct from this study) as a bait and successfully co-isolate the complete AMBRA1 (WD40)-DDB1-Cullin4-RBX1 complex, accessory protein DDA1 and a number of potential interactors involved in ubiquitination, cell cycle control, DNA-damage and stress response (see below). It is important to note that number of these proteins were identified in a previous study using full-length AMBRA1 (PMID 25499913). While these results still require an extensive statistical and biochemical verification, one may conclude that many of the AMBRA1 functional interactions are mediated by the globular core of WD40 domain. Further studies will be required to identify and characterize another set of regulatory interactions and IDRs. The affinity purification-based approaches for interactome mapping are challenging to apply for low-affinity or transient protein-protein interactions, and we are currently investigating alternative approaches. We hope to be able to provide a more comprehensive proteomics study and detailed mapping of different AMBRA1 interacting regions in the near future. In the meantime, we are happy to share a detailed list of interacting proteins and experimental conditions upon request.

AMBRA1^{WD40} associated proteins and their cellular functions. Colored ellipses represent different biological functions, including ubiquitination, cell cycle, DNA damage. In addition, tubulin and set of chaperone proteins copurify with AMBRA1^{WD40}. The interaction network figure was created using Cytoscape 3.10.

Furthermore, I still find the title "Structural basis for substrate recruitment by AMBRA1 E3 ligase receptor" somehow misleading, as it implies the characterization of a universal binding mechanism. At variance with that, the authors focus only on the structural characterization of the DDB1-AMBRA1 binding (not novel).

Point taken. We have changed the title.

Moreover, the required and performed additional experiments yield inconsistent results. Although the authors carried out experiments with single mutants, as suggested, the mutants V10R and L13R disrupt the binding with DDB1, while mutants W14R and H929A still exhibit interaction (Fig S9a). It remains unclear why single mutants show an impact on

Ubiquitination experiments (S9b) and on functional assays (Fig 4d), and this discrepancy is not addressed at all in the manuscript.

We have now extended our discussion concerning W14R and H929A mutants in lines 352 – 361.

Additionally, despite my suggestion to conduct more accurate experiments on autophagy, the authors have instead completely removed this section and focused only on a minimal set of experiments related to cell cycle. As for those, the authors included a preliminary analysis of the cell cycle phase distribution, which also misses a proper control (i.e., cells not interfered for AMBRA1 – Fig. 4d). The authors have also ignored the request to investigate the effects on genomic stability and the potential effect on CHK1 inhibition sensitivity, which is a significant oversight.

We have now included the siRNA control in Figure 4b. This control was present as a separate panel in the previous iterations of this figure but was not adequately explained and we apologize for the confusion.

We have now completed the DNA damage experiments in the presence of CHK1 inhibitor. The respective results are consistent with the cell cycle experiments (Figure 4c-d). We are sorry for not including all these results in the first round of revision. The initial setup of cell-based assays took a considerable amount of time.

Last, the majority of the new supplementary figures are inadequately prepared, lacking proper labeling for different panels, and exhibiting unexplanatory figure legends.

We thank the reviewers for pointing this out. We believe the insufficiencies in the supplementary figures legends and labeling are now corrected.